# Modelling the developmental spliceosomal craniofacial disorder Burn-McKeown syndrome using induced pluripotent stem cells

Katherine A. Wood[1,2], Charlie F. Rowlands[1,2], Huw B. Thomas[1], Steven Woods[3], Julieta O'Flaherty[3], Sofia Douzgou[1,2], Susan J. Kimber[3], William G. Newman[1,2], Raymond T. O'Keefe[1]*

1 Division of Evolution and Genomic Sciences, School of Biological Sciences, Faculty of Biology, Medicine and Health, The University of Manchester, Manchester, United Kingdom, 2 Manchester Centre for Genomic Medicine, Manchester Academic Health Science Centre, Manchester University NHS Foundation Trust, Manchester, United Kingdom, 3 Division of Cell Matrix Biology and Regenerative Medicine, School of Biological Sciences, Faculty of Biology, Medicine and Health, The University of Manchester, Manchester, United Kingdom

* rokeefe@manchester.ac.uk

**Data Availability Statement:** The datasets used and/or analysed during the current study can be found in the S1 File and S1 Data file. RNA-Seq

## Abstract

The craniofacial developmental disorder Burn-McKeown Syndrome (BMKS) is caused by biallelic variants in the pre-messenger RNA splicing factor gene *TXNL4A/DIB1*. The majority of affected individuals with BMKS have a 34 base pair deletion in the promoter region of one allele of *TXNL4A* combined with a loss-of-function variant on the other allele, resulting in reduced *TXNL4A* expression. However, it is unclear how reduced expression of this ubiquitously expressed spliceosome protein results in craniofacial defects during development. Here we reprogrammed peripheral mononuclear blood cells from a BMKS patient and her unaffected mother into induced pluripotent stem cells (iPSCs) and differentiated the iPSCs into induced neural crest cells (iNCCs), the key cell type required for correct craniofacial development. BMKS patient-derived iPSCs proliferated more slowly than both mother- and unrelated control-derived iPSCs, and RNA-Seq analysis revealed significant differences in gene expression and alternative splicing. Patient iPSCs displayed defective differentiation into iNCCs compared to maternal and unrelated control iPSCs, in particular a delay in undergoing an epithelial-to-mesenchymal transition (EMT). RNA-Seq analysis of differentiated iNCCs revealed widespread gene expression changes and mis-splicing in genes relevant to craniofacial and embryonic development that highlight a dampened response to WNT signalling, the key pathway activated during iNCC differentiation. Furthermore, we identified the mis-splicing of *TCF7L2* exon 4, a key gene in the WNT pathway, as a potential cause of the downregulated WNT response in patient cells. Additionally, mis-spliced genes shared common sequence properties such as length, branch point to 3' splice site (BPS-3'SS) distance and splice site strengths, suggesting that splicing of particular subsets of genes is particularly sensitive to changes in *TXNL4A* expression. Together, these data provide the first

datasets have been deposited with the Gene Expression Omnibus (GEO) under accession number GSE149369.

**Funding:** Funding: The research of WGN and RTO was supported by the Biotechnology and Biological Sciences Research Council (BB/N000258/1) (bbsrc.ukri.org). WGN was supported by National Institute for Health Manchester Biomedical Research Centre funding (IS-BRC-1215-20007) (manchesterbrc.nihr.ac.uk). KAW was supported by a Medical Research Council studentship (1916606) (mrc.ukri.org). CFR was supported by a Medical Research Council industrial Collaborative Awards in Science and Engineering (iCASE) studentship (1926882) with QIAGEN (mrc.ukri.org). SW and SJK were supported by the European Union SYBIL European Community's Seventh Framework Programme (FP7/2007-2013, 60230) (ec.europa.eu). The funders had no role in study design, data collection and analysis, decision to publish, or preparation of the manuscript.

**Competing interests:** The authors have declared that no competing interests exist.

insight into how reduced *TXNL4A* expression in BMKS patients might compromise splicing and NCC function, resulting in defective craniofacial development in the embryo.

## Introduction

Burn-McKeown syndrome (BMKS; OMIM 608572) is a rare craniofacial developmental disorder. There are currently fewer than 20 reported affected families with BMKS worldwide. The primary phenotype associated with BMKS is choanal atresia, observed in all patients to date. Additional phenotypic features include conductive and sensorineural hearing loss and typical craniofacial dysmorphic features, including cleft lip and/or palate, short palpebral fissures, lower eyelid coloboma, a short philtrum, a prominent nose with high nasal bridge, and large protruding ears, while visceral malformations such as congenital heart defects are also sometimes observed [1–6]. Intellectual development is usually unimpaired, although at least one reported BMKS patient suffers severe intellectual disability and developmental delay [7].

In 2014, Wieczorek et al. reported genetic variants in *TXNL4A* as causative in BMKS [4]. Most BMKS patients identified thus far have a 34bp deletion (chr18: g.77,748,581_77,748,614del (GRCh37, hg19), type 1 Δ34bp) within the promoter region of *TXNL4A* on one allele combined with a loss-of-function variant on the other allele. Different loss-of-function variants have been reported, including microdeletions, splice site, nonsense and frameshift variants [4,5]. Some individuals with BMKS do not have a compound heterozygous genotype, but are homozygous for a slightly different *TXNL4A* 34bp promoter deletion (chr18: g.77,748,604_77,748,637 (GRCh37, hg19), type 2 Δ34bp) [4,5,8]. Type 1 Δ34bp and type 2 Δ34bp promoter deletions reduced reporter gene expression by 59% and 72% respectively [4]. The more severe reduction in expression caused by the type 2 Δ34bp might explain why a homozygous type 2 Δ34bp is sufficient to cause BMKS, whereas a type 1 Δ34bp must be combined with a null allele to produce a BMKS phenotype. It is postulated that the BMKS phenotype is the result of a specific dosage of *TXNL4A*, where biallelic null variants in *TXNL4A* are likely to be incompatible with life.

*TXNL4A* encodes a component of the U5 snRNP, one of the core building blocks of the spliceosome, the macromolecular machine responsible for the splicing of pre-mRNAs [9–11]. The *S. cerevisiae* ortholog of *TXNL4A*, *DIB1*, encodes a small, highly conserved protein which is essential for pre-mRNA splicing, while null mutations in *DIB1* in *S. cerevisiae*, *S. pombe* and *C. elegans* are incompatible with life [4,12–16]. It has been postulated that Dib1 prevents premature spliceosome activation, with the departure of Dib1 defining the transition from the B complex to the B^act complex during the splicing cycle [16]. Reducing *DIB1* expression in *S. cerevisiae* led to defective assembly of the U4/U6.U5 tri-snRNP [4]. Given the high homology between *DIB1* and *TXNL4A*, it is likely that reduced *TXNL4A* expression resulting from the *TXNL4A* mutations observed in BMKS patients leads to reduced assembly of the human tri-snRNP, which in turn may affect the splicing of a specific subset of pre-mRNAs important in craniofacial development.

BMKS is one of five developmental craniofacial disorders caused by variants in core spliceosome components [6,17]. Given the universality of pre-mRNA splicing in the processing of all human pre-mRNAs, the very specific and tissue-restricted craniofacial phenotypes of these disorders are remarkable. In contrast, variants in other spliceosome components, some within the same spliceosome complex, are associated with a very clinically different phenotype, autosomal dominant retinitis pigmentosa [18–21]. Notably, variants in the splicing factor *CWC27*

have now been identified in individuals with retinitis pigmentosa, craniofacial defects, developmental delay and brachydactyly, showing that overlap of distinct disease phenotypes is possible [22].

Craniofacial development is an exquisitely complex process occurring largely during the first trimester of human gestation. The critical cells in embryonic craniofacial development are neural crest cells (NCCs), a transient cell population originating from the neuroectoderm located at the neural plate border during neurulation [23,24]. NCCs have stem-like properties and undergo some of the most extensive movements of any cell type during embryonic development [25]. NCCs form the derivatives of numerous differentiated cell and tissue types, including the dorsal root ganglia, peripheral nerves, pigment and adipose cells, and craniofacial tissues [26]. Cranial NCCs contribute to the patterning of the craniofacial skeleton. NCCs have an important role in patterning of the head, generation of the meninges and forebrain vasculature, and are required for proper brain patterning and growth [27]. *In vivo*, induction of NCCs relies on WNT, BMP, FGF, Activin/TGF-β and Notch/Delta signalling pathways [28–33]. Given the very significant role of NCCs in the development of craniofacial structures, a number of craniofacial disorders have been associated with defects in NCC biology including Treacher Collins syndrome, CHARGE syndrome and the spliceosomal craniofacial disorders Nager syndrome (NS) and Richieri-Costa Pereira Syndrome (RCPS) [6,17,24,34–39]. Therefore, it is likely that variants in *TXNL4A* in BMKS lead to defects particularly affecting NCCs, resulting in a craniofacial phenotype. However, to date, there have been no reports investigating the effects of BMKS-specific *TXNL4A* variants on NCC populations.

In this study, we generated induced pluripotent stem cell (iPSC) lines from a patient with BMKS (BMKS005 in [4,7]) and her unaffected mother. Patient iPSCs displayed slow proliferation and RNA-Seq analysis revealed a distinct subset of mis-expressed and mis-spliced pre-mRNAs. Differentiation of iPSCs to induced NCCs (iNCCs) followed by RNA-Seq analysis revealed defective iNCC differentiation in BMKS patient cells, likely arising from a diminished response to WNT signalling. RNA-Seq analysis revealed transcriptome-wide mis-expression and mis-splicing, in particular aberrant exon skipping, affecting distinct subsets of pre-mRNAs in patient iNCCs, with an enrichment for genes involved in NCC specification and function, such as the epithelial-to-mesenchymal transition (EMT). The aberrantly skipped exons shared common sequence properties, such as longer upstream and downstream intron length, and weaker splice acceptor and splice donor strengths, suggesting that related pre-mRNAs are most vulnerable to reduced *TXNL4A* expression. Taken together, these results provide evidence that BMKS-specific *TXNL4A* variants influence human NCC function and behaviour, and suggest how reduced *TXNL4A* expression leads to craniofacial defects.

## Materials and methods

### Ethical approval

The affected individual (BMKS005 in [4,7]) provided written informed consent in accordance with local regulations. Ethical approval for this study was granted by the National Research Ethics Service (NRE) (11/H1003/3, IRAS 64321) at The University of Manchester.

### Isolation of Peripheral Blood Mononuclear Cells (PBMCs) from whole blood

Whole blood samples were provided by the referring clinician in BD Vacutainer EDTA tubes (BD). Blood samples were stored at room temperature prior to PBMC isolation and PBMCs were isolated no more than 48h following blood collection. Blood samples were diluted in

phosphate-buffered saline (PBS) (Sigma) and layered over Ficoll-Paque (GE Healthcare), followed by density-gradient centrifugation at 400g for 40 min at room temperature. The PBMCs were transferred to a new tube, diluted with PBS and centrifuged at 200g for 10min at room temperature. The cell pellet was washed once by centrifugation in PBS. The PBMCs were either resuspended immediately in complete Erythroid Expansion Medium (StemSpan-SFEM II with 100X StemSpan Erythroid Expansion Supplement) (Stem Cell Technologies) for reprogramming or resuspended at a density of 0.5 x $10^6$ PBMCs per cryo vial in PSC cryomedium (Gibco) for long-term storage in liquid nitrogen.

## Reprogramming of PBMCs to iPSCs

Approximately 50,000 PBMCs were resuspended in complete Erythroid Medium (StemSpan-SFEM II with 100X StemSpan Erythroid Expansion Supplement) (Stem Cell Technologies) and transferred to a single well of a 6-well tissue culture plate for 24h. All non-adherent cells were transferred to a new well of the 6-well plate and incubated for a further 24h in the same medium. Approximately 75% of the medium was removed and replaced with 1.5ml fresh complete Erythroid Medium every 48h for a further six days. Following this expansion, 50,000 cells were transferred to a 15ml Falcon tube and centrifuged at 300g for 5min at room temperature. Cells were then transduced using CytoTune-iPS 2.0 Sendai virus (Invitrogen) as outlined in the supplier's protocol at an MOI of 5:5:3 (KOS [Klf4, Oct4 and Sox2] = 5: hc-Myc = 5: hKlf4 = 3) in 250µl complete Erythroid Medium and centrifuged at 300g for 35min at room temperature. Following centrifugation, 150µl complete Erythroid Medium was added to give 400µl in total and plated (200µl/well) in two wells of a standard 24-well tissue culture plate (day 0). 200µl complete Erythroid Medium was added to each well on day 1, 3 and 4 following transduction (without removal of previous medium on these days). On day 5 after transduction, the cells and medium were transferred to fresh vitronectin (VTN)-coated 6-well tissue culture plates (10µl truncated vitronectin recombinant human protein (Invitrogen) + 1ml PBS (5µg/ml VTN) to each well for 1h at room temperature) containing 0.5ml complete Repro-TeSR medium (ReproTeSR basal medium with 25X ReproTeSR supplement) (Stem Cell Technologies). After 24h incubation, 1ml complete ReproTeSR was added (without removing previous medium from the wells), followed by daily removal and replacement of medium with complete ReproTeSR. After a minimum of 10 days post-transfer to VTN-coated plates, emerging iPSC colonies that were large enough to isolate were manually picked using a sterile pulled glass pipette into VTN-coated 6-well tissue culture plates containing complete TeSR-E8 media (TeSR-E8 basal medium with 25X TeSR-E8 supplement) (Stem Cell Technologies) using a SMZ1000 Nikon dissecting microscope (Nikon) to form clonal lines at passage 1 and incubated and fed daily with complete TeSR-E8 for 7–10 days. After this time period, the colonies were passaged to passage 2 using 0.5mM UltraPure ethylenediaminetetraacetic acid (EDTA) (Invitrogen) and clonal lines were cultured as iPSCs thereafter (see below). All clonal lines were taken to passage 15 prior to any characterisation or further experimentation to avoid the instability of early passage iPSC cultures and to ensure colonies were no longer supported by viral vector gene expression.

## Culture of iPSCs

Pluripotent iPSCs were cultured as colonies in feeder-free conditions at 37˚C with 5% $CO_2$ on VTN-coated 6-well tissue plates in TeSR-E8 with daily feeding. Colonies were passaged every 4–7 days using 3 – 5min incubation with 0.5mM UltraPure EDTA (Invitrogen) as required.

## Genotyping

iPSC colonies were cultured in 6-well tissue culture plates until colonies were almost touching. Genomic DNA (gDNA) was extracted from iPSC colonies using the Qiagen QIAamp DNA Mini Kit. Genotyping was performed using a PCR method. gDNA was amplified using Phusion High-Fidelity DNA Polymerase (New England Biolabs, NEB) according to manufacturer's recommendations and primer pairs listed in **S1 Table in S2 File**. PCR products were separated by agarose gel electrophoresis supplemented with SafeView nucleic acid stain (NBS Biologicals) and imaged on a UV transilluminator. NEB 1kb Plus DNA Ladder was loaded alongside PCR products as a DNA size standard. PCR products were purified using the Pure-Link Quick Gel Extraction Kit (Invitrogen) and the identities confirmed by Sanger sequencing performed by Eurofins Genomics.

## RNA isolation

Total RNA was extracted from cells using TRIzol reagent (Invitrogen) according to the manufacturer's protocol. The resulting RNA was cleaned up using the Qiagen RNeasy Mini RNA Isolation Kit, including an on-column DNA digestion step. The RNA was either used immediately to synthesise cDNA or stored at -80˚C.

## cDNA synthesis

Total RNA was converted into cDNA using Superscript IV (Invitrogen) and random hexamers (Invitrogen) according to supplier's instructions. The cDNA was either used immediately as a template for quantitative or reverse transcription PCR or stored at -20˚C.

## Quantitative PCR (qPCR)

qPCR reactions were performed with 2X PowerUp SYBR Green PCR Master Mix (Invitrogen). Primer pairs were either retrieved from Primer Bank (https://pga.mgh.harvard.edu/primerbank/), designed manually or retrieved from the relevant literature, and are listed in **S1 Table in S2 File** [40,41]. Fluorescence was detected using the StepOnePlus Real-Time PCR System (Applied Biosystems) in 96-well plate format, using the standard comparative $C_T$ reaction cycle programme and associated data analysis software. Gene expression was assessed using the $\Delta\Delta C_T$ method, using *ACTB* as a stable endogenous control gene. *ACTB* was chose as the endogenous control gene because *ACTB* expression was not significantly different between any of the cell lines from the RNA-Seq datasets presented here and the $C_T$ values were similar to the $C_T$ values for the test genes [42]. Specificity of qPCR primers was confirmed by a single peak in the melt curve analysis included in the standard qPCR reaction cycle programme.

## Reverse Transcription PCR (RT-PCR)

RT-PCR reactions were performed with Phusion High-Fidelity DNA Polymerase (NEB) according to manufacturer's recommendations, using cDNA as a template. Primer pairs were designed using Primer-BLAST (https://www.ncbi.nlm.nih.gov/tools/primer-blast/) and are listed in **S1 Table in S2 File** [43]. PCR products were separated by agarose gel electrophoresis supplemented with SafeView nucleic acid stain (NBS Biologicals) and imaged under a UV transilluminator. NEB 1kb Plus DNA Ladder was loaded alongside PCR products as a DNA size standard. Where stated, RT-PCR products were purified using the PureLink Quick Gel Extraction Kit (Invitrogen) and identities confirmed by Sanger sequencing by Eurofins Genomics.

## Immunostaining

Three to five colonies of established iPSCs (passage 15 or higher) were seeded in VTN-coated 24-well tissue culture plates and incubated for 48h. Cells were fixed with 4% paraformaldehyde (PFA) in PBS for 10min at room temperature, followed by three washes with PBS. For intracellular markers, PBS was supplemented with 0.1% Triton X-100 (PBT). Fixed cells were blocked for 30min at room temperature in PBS or PBT supplemented with 10% goat serum with 0.05% Tween-20 and 0.1% bovine serum albumin (BSA). Blocked cells were incubated overnight at 4˚C with primary antibodies (and appropriate serotype controls) (**S2 Table in S2 File**) diluted in PBS or PBT supplemented with 1% goat serum. The cells were washed three times with PBS, then incubated for 2h at room temperature with Alexa Fluor secondary antibodies (**S2 Table in S2 File**) and nuclear staining with 4',6-diamidino-2-phenylindole (DAPI) (Invitrogen). Images were captured on a BX51 fluorescence microscope (Olympus) with a Retiga-SRV camera (QImaging). Image analysis was performed using ImageJ and the QCapture Pro (QImaging) software [44].

## Karyotyping

Karyotyping of pluripotent iPSCs was conducted by the North West Genomic Laboratory Hub, Manchester Centre for Genomic Medicine.

## Apoptosis assays

Cell apoptosis was assessed by annexin-V staining. For assessment of iPSCs, iPSC colonies were grown to approximately 60% confluency on VTN-treated 6-well tissue culture plates. For iNCCs, 120h differentiated iNCCs cultured on VTN-treated 6-well tissue culture plates were used. Cells were disrupted using trypsin-EDTA (Sigma) treatment and cells pelleted by centrifugation. Cells were washed once in PBS and then washed once in 1X Annexin-V Binding Buffer (ABB) (Invitrogen). The cells were resuspended at a density of $1 \times 10^6$ cells/ml in 1X ABB, and 7.5μl APC-conjugated Annexin-V (Invitrogen) added to 100μl cell suspension. The cells were incubated for 20min at room temperature in the dark, washed in 1X ABB and resuspended in 200μl 1X ABB. 0.25μg/ml DAPI (Invitrogen) was added and the resulting samples immediately analysed by flow cytometry. Subpopulations were ascertained in a BD LSRFortessa flow cytometer (gated using an unstained control) as follows: live cells (annexin V-, DAPI-); necrotic cells (annexin-V-, DAPI+); early apoptotic (annexin V+, DAPI-); late apoptotic (annexin V+, DAPI+). Early and late apoptotic subpopulations were pooled for analysis.

## Proliferation assays

Cell proliferation was assessed using the Abcam MTT Cell Proliferation Assay Kit (Abcam) according to manufacturer's instructions. In brief, $5 \times 10^3$ cells/well were seeded in 96-well tissue culture plates in triplicate in TeSR-E8 medium supplemented with Revitacell (Gibco) (10μl Revitacell per 1ml medium) and incubated for 48h. To quantify metabolically active cells, medium was replaced with a 50:50 mix of MTT reagent and TeSR-E8 supplemented with Revitacell and cells incubated for 3h at 37˚C. MTT solvent was added to each well and the plate shaken in the dark on an orbital shaker for 15min. Absorbance at 595nm was immediately assessed spectrophotometrically.

## RNA-Seq

Total RNA was isolated from cells as described above. For RNA-Seq analysis of iPSCs, iPSC colonies were grown until almost confluent before RNA isolation. For RNA-Seq analysis of iNCCs, RNA was isolated from iNCCs 120h after initiation of differentiation. 1μg of RNA

from each sample was diluted in 20μl nuclease-free water, and the quality and integrity of the RNA samples assessed using a 2200 TapeStation (Agilent Technologies), with RNA Integrity Number (RIN) ≥7.5 as the minimum quality cut-off for RNA samples. A poly-A enrichment library was then generated using the TruSeq Stranded mRNA assay (Illumina) according to manufacturer's protocol, and 76bp (+ indices) paired-end sequencing carried out on an Illumina HiSeq 4000 instrument. Output data were de-multiplexed (allowing one mismatch) and BCL-to-Fastq conversion performed using Illumina's bcl2fastq software, version 2.17.1.14. Low quality bases and adaptor sequences were trimmed using Trimmomatic, and reads aligned to the GRCh38 genome assembly using the single-pass mode of the STAR aligner (v2.5.3a), as well as to the transcriptome according to the Gencode v28 human gene annotation (downloaded from http://www.gencodegenes.org/human/release_28.html) [45,46]. Differential expression analysis was conducted using the Bioconductor DESeq2 package v1.20.0 using default parameters [47].

rMATS v.4.0.2 was used to identify mis-splicing events [48–50]. Results were filtered to retain only events with a p-value < 0.05 and FDR < 0.05. For each gene showing a skipped exon (SE) event, control sequences were generated from elsewhere within the gene (internal controls) of exon-intron-exon-intron-exon (E-I-E-I-E). These were generated randomly for each gene from the longest protein-coding transcript with transcript support level in the Gencode v28 annotation (where protein-coding transcripts were present); these sequences could not overlap the loci of any of the corresponding event E-I-E-I-Es from the original rMATS data, making construct generation impossible for some transcripts with low exon number. A single construct was chosen randomly from the pool of valid constructs for each transcript to make the final control sets. To generate external control sequence constructs, expression analysis of the iNCC RNA-Seq dataset was first carried out with RSEM v1.3.0 using default parameters [51]. To generate control constructs for which the absence of observed effects on splicing was not due to insufficient expression, we selected the 2000 most expressed genes that were not located on the mitochondrial genome, not present in any of the five filtered rMATS datasets, and had a corresponding transcript (with matching Ensembl transcript ID and version number) in the Ensembl MANE v0.7 curated transcript set. Random E-I-E-I-E constructs were then generated for each of these 2000 transcripts.

GO analysis was performed using the PANTHER Overrepresentation Test (released 11-07-2019) (Fisher's Exact test, FDR correction) with the GO database released 08-10-2019, and annotation data set GO Biological Processes Complete via the Gene Ontology Resource Consortium webserver (http://geneontology.org/) [52,53]. Analysis of the associated networks and upstream regulators was performed using Ingenuity Pathway Analysis (IPA) (https://digitalinsights.qiagen.com/products-overview/discovery-insights-portfolio/content-exploration-and-databases/qiagen-ipa/) [54]. 5' and 3' splice site strengths were quantified using the webserver of MaxEntScan::score5ss (http://hollywood.mit.edu/burgelab/maxent/Xmaxentscan_scoreseq.html) and MaxEntScan::score3ss (http://hollywood.mit.edu/burgelab/maxent/Xmaxentscan_scoreseq_acc.html), respectively [55]. Branch point sequence analysis was performed using the webserver of the SVM-BPfinder tool (http://regulatorygenomics.upf.edu/Software/SVM_BP/) [56].

## Differentiation

Differentiation of iPSCs to iNCCs was performed following a modified version of the protocol described by Leung et al. (2016) and implemented by Barrell et al. (2019) [32,33]. iPSC colonies were cultured in VTN-coated 6-well tissue culture plates until almost touching. Colonies were disrupted using EDTA and the cells resuspended in Neural Crest Induction (NCI) medium (50:50 DMEM:F12 (Sigma) supplemented with 1X Glutamax (Gibco), 0.5% BSA

(Sigma) and 30μM CHIR 99021 (Sigma)) supplemented with 10μM Y-27632 (Sigma), a ROCK inhibitor, with pipetting 15–20 times to generate a single-cell suspension. $2 \times 10^5$ cells/well were plated in VTN-coated 6-well tissue culture plates and incubated in NCI medium + Y-27632 for 48h with daily media replenishment. The cells were then incubated in NCI medium without Y-27632 for a further 72h with daily media changes. After the total 120h, differentiated iNCCs were either used for RNA isolation or further experimentation.

## Cell migration

iNCCs were differentiated for 144h in a 6-well tissue culture plate until 90–100% confluent in a monolayer. A P1000 pipette tip was used to make a straight scratch through the monolayer. Debris was removed by washing with PBS twice, and the medium replaced with fresh NCI medium. The tissue culture plate was transferred to the sterile tissue culture chamber of a live imaging Nikon Eclipse Ti Inverted Microscope at 37˚C with 5% $CO_2$ and cell migration monitored by imaging every 1h for 24h. Images were analysed using the MRI wound healing tool in Fiji [57].

## Statistical analysis

Graph drawing and statistical analysis was performed using GraphPad Prism version 8. Kolmogorov-Smirnov tests were used to test for normality. Parametric Student's t-tests were used to assess statistical significance of the difference between two means. Non-parametric Mann-Whitney tests were used to assess statistical significance of the difference between two medians. * $p$-value < 0.05, ** $p$-value < 0.01, *** $p$-value < 0.001, **** $p$-value < 0.001.

## Results

### Derivation of iPSC clonal lines from a patient with BMKS

One challenge in studying developmental craniofacial disorders is finding a suitable and relevant model. While developing human embryo craniofacial tissue is the most relevant tissue to investigate, craniofacial defects may not be present until birth and limitations on such research preclude this as a viable route. We therefore aimed to establish clonal induced pluripotent stem cell (iPSC) lines from a family with a BMKS affected individual as an *in vitro* human cell model for disease mechanisms underlying BMKS. Whole blood samples were collected from a female patient with BMKS and her parents recruited in the United Kingdom. The patient was 11 years of age at the time of blood collection. The patient is the fourth and youngest child of unaffected consanguineous parents of Pakistani origin; none of her siblings are affected. The clinical features of the patient have been documented previously [4,7]. Unlike all other patients with BMKS described to date, the patient displays developmental delay and severe intellectual disability; however, whole-exome sequencing has not revealed any genetic or structural/copy-number variants that could account for this significant cognitive impairment [7]. It is unclear whether her intellectual disability represents a novel feature of the clinical presentation of BMKS which would be more apparent if there was a larger cohort of patients.

The genotypes of the patient and her parents were previously reported by Wieczorek et al. [4]. The patient has a type 1 34bp deletion (type 1 Δ34bp) in the promoter of *TXNL4A* on one allele, inherited from her father, in combination with a 1bp deletion (chr18: g.77,748,262delA, RefSeq NM_006701.2; c.131delT (p.Val44Alafs*48)) in exon 1 of *TXNL4A* on the other allele, inherited from her mother (**Table 1**). The 1bp deletion leads to a frameshift in the coding sequence; however, the resulting premature termination codon (PTC) (codon 92) is included in the final exon (exon 3) of *TXNL4A* and thus transcripts from this allele are thought to evade nonsense-mediate decay (NMD). However, the protein encoded by these mutant transcripts is

**Table 1. Properties of all iPSC lines used in this study.**

| Individual | Sex | Disease status | iPSC lines | *TXNL4A* allele 1 genotype | *TXNL4A* allele 2 genotype | iPSC Karyotype |
|---|---|---|---|---|---|---|
| Patient | F | Burn-McKeown syndrome with intellectual disability | KW181A, KW181B | chr18: g.77,748,262delA, RefSeq NM_006701.2; c.131delT (p. Val44Alafs*48) | Promoter type 1 Δ34bp | 46XX, normal |
| Mother | F | Unaffected | KW191A, KW191B | chr18: g.77,748,262delA, RefSeq NM_006701.2; c.131delT (p. Val44Alafs*48) | Wildtype | 46XX, normal |
| Father | M | Unaffected | NA | Wildtype | Promoter type 1 Δ34bp | NA |
| Unrelated control 1 | M | Unaffected | SW162C | Wildtype | Wildtype | 46XY, normal (1 full and.1 partial karyotype only due to poor sample quality) |
| Unrelated control 2 | M | Unaffected | SW171A | Wildtype | Wildtype | 46XY, normal |

Summary of the iPSC lines generated and/or characterised in this study along with phenotype information for the corresponding individuals from which the iPSCs were derived. Genomic coordinates are described using the GRCh37/hg19 reference genome.

presumably non-functional because of the shift in the reading frame, rendering the mutation as a loss-of-function variant [4].

Peripheral blood mononuclear cells (PBMCs) were isolated from whole blood and reprogrammed to iPSCs. Two established iPSC clonal lines were generated for the patient and her mother, but attempts to reprogram PBMCs from the patient's father were unsuccessful. Additionally, we obtained two unrelated unaffected control iPSC lines from healthy individuals (**Table 1**). After reprogramming, all iPSC lines displayed pluripotent stem cell-like morphology (growing as compact colonies with distinct borders and well-defined edges, with cells having large nuclei, prominent nucleoli and high nucleus:cytoplasm ratio), had high mRNA expression levels of the pluripotency marker *OCT3/4*, *NANOG* and *SOX2* transcripts, stained positively for a number of pluripotency markers and negatively for SSEA-1, a marker of early differentiation, and all iPSC lines had a normal karyotype (**S1A-S1C Fig in S2 File**). Furthermore, genotyping confirmed the expected compound heterozygous *TXNL4A* variants in the patient iPSC lines, the heterozygous *TXNL4A* loss-of-function 1bp deletion in the mother iPSC lines, and wildtype *TXNL4A* in the unrelated control lines (**Table 1**, **S1D-S1E Fig in S2 File**). The expression of *TXNL4A* was significantly lower in patient iPSCs (which contain the type 1 Δ34bp) compared to iPSCs derived from her mother and control individuals without the type 1 Δ34bp mutation (**Fig 1A**) [4]. Interestingly, there were no significant differences in *TXNL4A* expression between iPSCs generated from the mother and those from the unrelated control individuals (**Fig 1A**). As discussed, although the heterozygous 1bp *TXNL4A* deletion found in both the patient and her mother induces a frameshift and PTC in the open reading frame, this PTC is present in the final exon meaning that transcripts containing the mutation most likely evade NMD and will still be detected by qPCR. However, while the maternal iPSC lines appeared to have unchanged overall *TXNL4A* mRNA expression levels, transcripts from the mutant allele will carry the frameshift and so encode a non-functional protein.

## BMKS patient iPSCs proliferate slowly but are not more apoptotic than control lines

Alterations in proliferation, the cell cycle, apoptosis and changes in the transcriptome have been observed in models of other spliceosomal developmental craniofacial disorders [24,58–

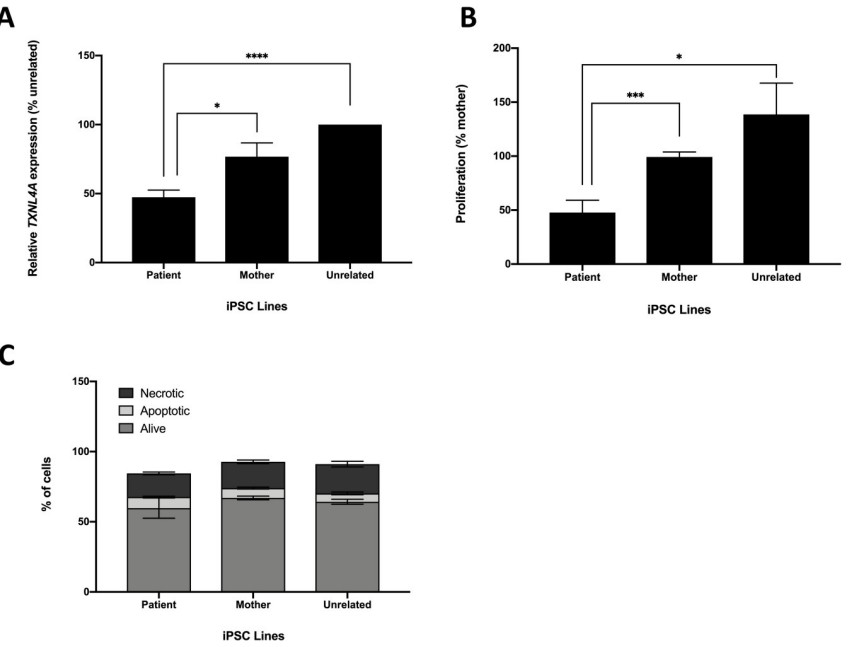

**Fig 1. Characteristics of BMKS patient-, mother- and unrelated control-derived iPSCs.** For the analyses, data from two patient iPSC lines, two parent iPSC lines and two unrelated control iPSC lines were pooled. A) Relative *TXNL4A* mRNA expression levels for patient iPSCs compared to mother iPSCs and unrelated control iPSCs, determined using qPCR of cDNA from each cell line. Graphs were obtained using the $\Delta\Delta C_T$ method with *ACTB* as the endogenous reference gene and normalised to the unrelated control line SW171A. n = 4. B) Relative proliferation for patient iPSCs compared to mother iPSCs and unrelated control iPSCs using an MTT assay to monitor cell proliferation, normalised to the KW191A maternal line. n = 4. C) Percentages of apoptotic, necrotic and live cells for patient, mother and unrelated control iPSCs determined using co-staining with annexin-V and DAPI then quantification of cell subpopulations by flow cytometry. n = 3. * p-value < 0.05, ** p-value < 0.01, *** p-value < 0.001, **** p-value < 0.0001.

61]. We therefore set out to identify whether any of these cellular properties were altered in our BMKS patient-derived iPSC lines. Although NCCs are the most disease-relevant cell type, given the importance of pre-mRNA splicing in all cell types it was expected that differences in cell function and behaviour would be observed even at the pluripotent stage.

During routine culturing of the iPSC lines, the patient lines required less frequent passaging than iPSC lines derived from healthy individuals. Given the importance of *TXNL4A* in the spliceosome and of pre-mRNA splicing in all aspects of cell function and behaviour, it was very likely that reduced expression of this core pre-mRNA splicing factor would compromise in some way the ability of cells to grow and/or proliferate, as with other models of human splicing factor knockdown or inhibition [58,62]. We therefore performed a 3-(4,5-dimethylthiazol-2-yl)-2,5-diphenyltetrazolium bromide (MTT) tetrazolium assay to quantify any differences in proliferative rate between the iPSC lines. Proliferation was significantly lower in patient iPSC lines compared to both mother and unrelated control lines (**Fig 1B**). There was no significant difference in proliferation between mother and unrelated control lines (**Fig 1B**). We also investigated apoptosis in the iPSC lines using annexin-V staining, and co-staining with DAPI as a marker of necrotic cells. Subpopulations of stained cells were quantified using flow cytometry. However, there were no significant differences in the numbers of apoptotic or necrotic cells between the patient, mother and unrelated control iPSC lines (**Fig 1C**).

## Widespread changes in gene expression in BMKS patient iPSCs

To quantify differential gene expression and mis-splicing events in BMKS iPSCs resulting from reduced *TXNL4A* transcription, RNA-Seq analysis was performed on all six clonal iPSC lines. Triplicate RNA samples were isolated from the iPSCs (passage 20–25) cultured under normal conditions and analysed by RNA-Seq. Differential gene expression analysis was performed using DESeq2, while differential splicing events were analysed using rMATS. For all bioinformatic analysis, only events that were significantly different in both patient lines in relation to pooled maternal lines and both control lines together (i.e. events that were significantly different in the patient versus all others) were considered as differential events likely resulting from the patient-specific genotype. However, validations were also performed between the patient and the maternal or control lines separately. Principal Component Analysis (PCA) revealed close clustering of the patient and maternal lines compared to the two unrelated control lines (**Fig 2A**).

In total, 1181 genes (both protein-coding and non-coding) were differentially expressed between patient and pooled control iPSC lines (p < 0.05) (**Fig 2B**, **S1 File**). There were 346 genes (29%) that displayed significantly increased expression in patient iPSC lines while 835

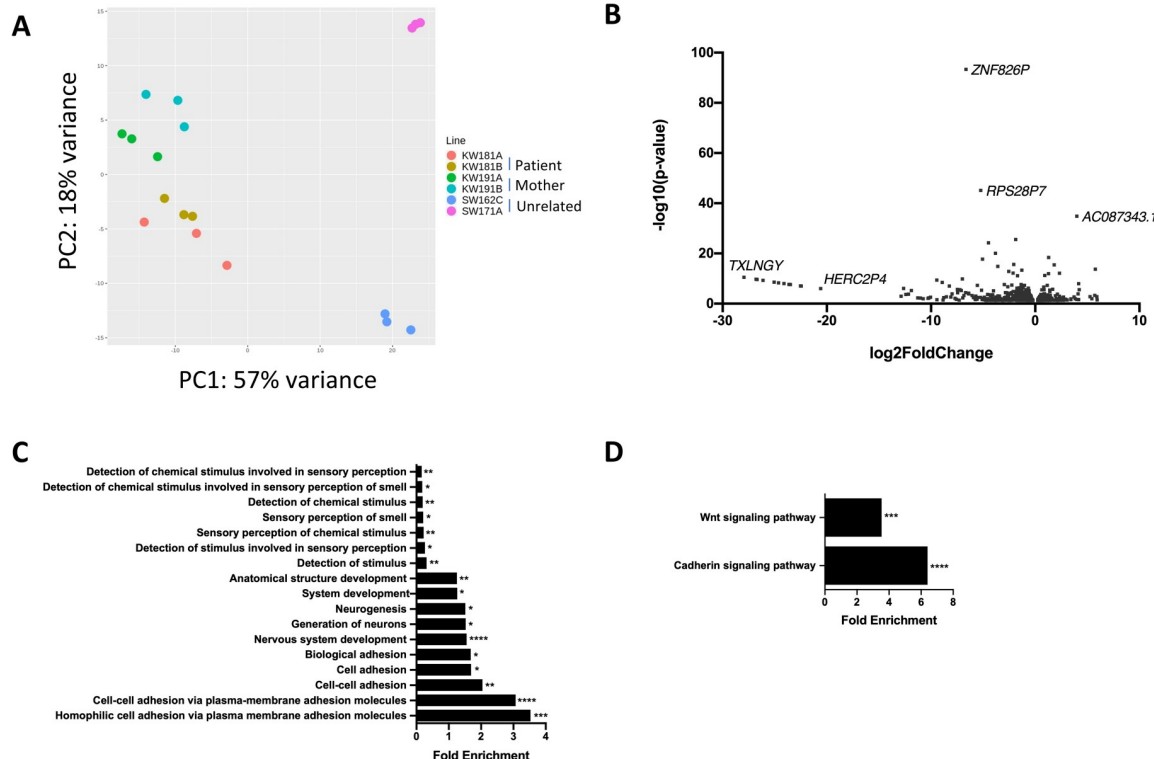

**Fig 2. Whole transcriptome RNA-Seq analysis of iPSCs.** RNA-Seq experiments were performed on triplicate samples of RNA isolated from all six iPSC lines (KW181A and KW181B = patient iPSC lines; KW191A and KW191B = mother iPSC lines; SW162C and SW171A = unrelated control lines). For analysis, data from the patient lines was pooled and compared with data pooled from both the mother and unrelated control lines. A) Principal Component Analysis (PCA) for all iPSC lines from RNA-Seq data. B) Volcano plot showing all 1181 significantly differentially expressed genes (DEGs) in patient iPSCs compared to pooled control iPSCs obtained using DESeq2 analysis of RNA-Seq data. Log2foldchange < 0 = significantly upregulated DEGs in patient iPSCs compared to pooled control iPSCs, log2foldchange > 0 = significantly downregulated DEGs in patient iPSCs compared to pooled control iPSCs. C) All enriched Gene Ontology (GO) terms for biological processes associated with DEGs in patient iPSCs compared to pooled control iPSCs, obtained using the PANTHER Overrepresentation Test for Biological Processes. D) Enriched GO terms for biological pathways associated with DEGs in patient iPSCs compared to pooled control iPSCs, obtained using the PANTHER Overrepresentation Test for Biological Pathways. * p-value < 0.05, ** p-value < 0.01, ***p-value < 0.001, **** p-value < 0.0001.

genes (71%) had significantly decreased expression, including *TXNL4A* (**S1 File**). No other significant change in splicing factor expression was observed in the patient iPSC lines except for an increase in core splicing factor *PRPF3* expression and a decrease in auxiliary splicing factor U2AF1 expression (**S3 Table in S2 File, S1 File**). To determine whether particular subsets of genes were differentially expressed, Gene Ontology (GO) enrichment analysis for biological processes was performed on all 1181 differentially expressed genes (DEGs). Among the most enriched GO terms were terms related to cell-cell adhesion via plasma membrane adhesion molecules, nervous system development and neurogenesis, and anatomical structural development (**Fig 2C**). Interestingly, GO enrichment for biological pathways identified WNT signalling and cadherin pathways enriched in the DEGs (**Fig 2D**). Cell-cell adhesion is very important in neural crest induction and differentiation, in particular during the EMT, delamination and subsequent migration [63–66]. Additionally, WNT signalling plays a critical role in the differentiation of NCCs [32,67,68].

Ingenuity Pathway Analysis (IPA) was then used to investigate the DEGs in terms of known canonical pathways, upstream regulators, networks, molecular and cellular functions and physiological system development and function (**S2 Fig in S2 File**). Among the most enriched upstream regulators were EOMES (p = $3.9 \times 10^{-5}$) and PAX7 (p = $5.9 \times 10^{-5}$) (**S2B Fig in S2 File**). EOMES is a critical developmental transcription factor which plays a role, among others, in brain development and specification and proliferation of cells of the cerebral cortex [69]. PAX7 is also a developmental transcription factor and an important marker of the neural border during neural crest induction [32,70]. Among the most enriched functional networks identified by IPA associated with the DEGs were networks involved in cellular development, cellular growth and proliferation, and developmental disorders, showing a prevalence for networks involved in embryonic development (**S2C Fig in S2 File**).

## Mis-splicing in BMKS iPSCs

rMATS was used to quantify and analyse differential splicing between patient and pooled control iPSCs from the RNA-Seq data. In total, rMATS identified 1511 differential splicing events (p <0.05, FDR < 0.05) in 1096 different genes, 75 of which were also differentially expressed (**Fig 3A, S1 File**). A number of these events were validated by RT-PCR (**S3 Fig in S2 File**). 96 genes displayed more than one type of mis-spicing event, and one gene (*SNHG14*) presented four types of mis-splicing while nine genes presented with three different forms of mis-splicing (**Fig 3B**, **S4 and S5 Tables in S2 File**). It is unclear why these particular genes are so vulnerable to multiple forms of mis-splicing. No significant change in the splicing of core or auxiliary spliceosome factors was observed in the patient iPSC lines (**S3 Table in S2 File, S1 File**).

To investigate whether particular subsets of genes were mis-spliced in the BMKS patient iPSCs, GO enrichment and IPA analysis was performed on all 1096 mis-spliced genes (**S4 Fig in S2 File**). There were no enriched GO terms with a greater than 0.5-fold enrichment. However, among the top upstream regulators identified by IPA were SMARCD3 and SMARCA4, both of which are important in neural progenitor differentiation, particularly in a switch from a stem cell-like fate to a post-mitotic differentiated state, which may be linked to the patient's intellectual disability (**S4B Fig in S2 File**). The top associated molecular and cellular functions with the mis-spliced genes identified by IPA included cell-to-cell signalling and interaction, cellular assembly and organisation, and cellular function and maintenance, all of which have an important role in embryonic and craniofacial development (**S4D Fig in S2 File**). Among the top associated networks identified by IPA were the cell cycle, and hereditary and developmental disorders, again revealing an enrichment for networks involved in development and disorder-related processes (**S4C Fig in S2 File**).

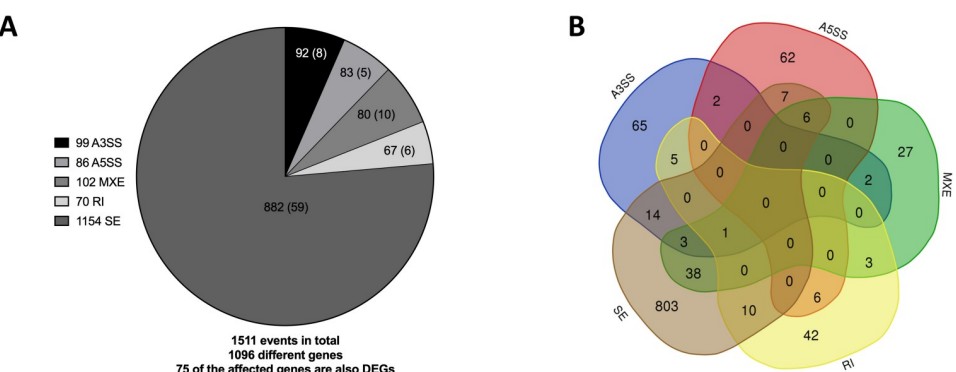

**Fig 3. Differential alternative splicing events in iPSCs.** A) Number of significantly different alternative 3' splice site (A3SS) and alternative 5' splice site (A5SS) events, mutually exclusive exon (MXE) events, intron retention events (RI) events and exon skipping (SE) events in pooled patient iPSCs compared to pooled maternal and unrelated control iPSCs, as determined by rMATS. The total number of events for each class of differential splicing is shown in the legend, the number of affected genes for each class of event shown in the corresponding segments of the pie chart and the number of the genes which are also differentially expressed for each class of event shown in brackets. B) Venn diagram showing the numbers of differentially spliced genes in patient iPSCs compared to pooled control iPSCs obtained by rMATs.

## Differentiation of BMKS patient iPSCs to iNCCs

Having investigated the effects of reduced *TXNL4A* expression in BMKS patient-derived iPSCs at the pluripotent stage, we turned our attention to differentiating the iPSCs to induced neural crest cells (iNCCs) as a more relevant, disorder-specific cell type. Here, we employed a protocol first described by Leung et al. (2016) which relies on the activation of WNT signalling [32]. WNT signalling is critical for NCC induction in a range of human and animal models, and this 120h protocol was designed to mimic normal neural crest differentiation *in vivo*. WNT signalling is activated via the inhibition of the glycogen synthase kinase-3 (GSK3), and within the first 72h there should be a gradual reduction in expression of pluripotency markers (*OCT3/4*, *SOX2* and *NANOG*) while the expression of neural border genes (*PAX3*, *PAX7*, *MSX1*, *MSX2*, *ZIC1* and *TFAP2A*) should increase by 72h. The expression of established neural crest marker genes such as *SOX10*, *SNAI2*, *NR2F1* and *FOXD3* (which is also a marker of pluri-potency, and so *FOXD3* expression initially decreases during the first 48h of differentiation) should be increased from approximately 96h onwards, indicating the acquisition of a defined neural crest fate. Furthermore, the activation of WNT signalling can be evidenced by a dra-matic and progressive increase in the expression of *AXIN2*, a canonical WNT target gene, over the course of the 120h differentiation protocol.

We initially optimised the differential protocol in the healthy unaffected control iPSC line SW171A (**Table 1**) to ensure robust differentiation and that the differentiated cells displayed the expected characteristics of iNCCs. Morphological analysis (**S5A Fig in S2 File**), qPCR anal-ysis of gene expression (**S5B-S5C Fig in S2 File**) and a wound healing assay showing the migratory ability of the differentiated cells (indicating successful delamination and an EMT) (**S5D Fig in S2 File**) confirmed the SW171A control cells had assumed a neural crest fate using the selected differentiation protocol. All six iPSC lines (**Table 1**) were then differentiated into iNCCs. Although the cells were plated at identical densities at the start of the iNCC differ-entiation protocol, fewer iNCCs were observed in the patient lines compared to the control lines at all time points examined after 0h. Annexin-V staining revealed no significant differ-ences in apoptosis between the patient iNCCs and the mother and unrelated control iNCCs, similar to the findings in the iPSCs (**Fig 4A**).

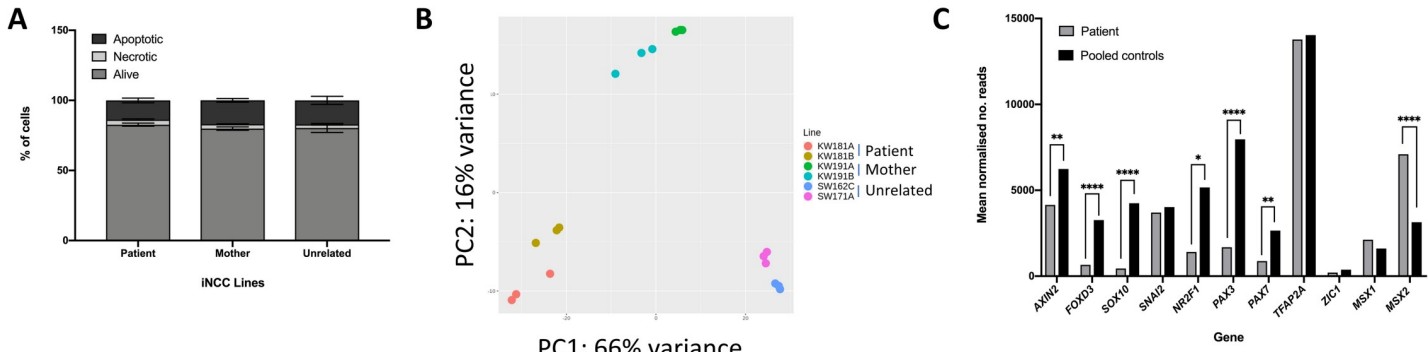

**Fig 4. Differentiation of iPSCs to iNCCs.** A) Percentages of apoptotic, necrotic and live cells for patient, mother and unrelated control iNCCs determined using co-staining with annexin-V and DAPI and quantification of cell subpopulations by flow cytometry. Data from the two patient lines, the two mother lines and the two unrelated control lines was pooled for analysis. n = 3. B) Principal Component Analysis (PCA) for all iPSC lines from RNA-Seq performed on all iNCC lines. Whole transcriptome RNA-Seq analysis was performed on triplicate samples of RNA isolated from all six iNCC lines. For analysis, data from the patient lines was pooled and data from the mother and unrelated control lines was pooled (pooled controls). C) Key marker gene expression levels in iNCCs from RNA-Seq data. Mean normalised read counts for *AXIN2*, neural crest and neural border marker genes. * p-value < 0.05, ** p-value < 0.01, *** p-value < 0.001, **** p-value < 0.0001.

To assess the extent of differentiation at the gene expression level between the iNCC cell lines, RNA-Seq analysis was performed in triplicate for 120h iNCCs from the patient, mother and unrelated control lines, and differential gene expression and differential splicing analysed between the lines. Data analysis and sample pooling was identical to the analysis performed for the iPSCs to allow a direct comparison. Interestingly, PCA analysis of the iNCCs revealed considerably less clustering between the patient and the maternal cells (**Fig 4B**) than the PCA analysis of the iPSCs (**Fig 2A**), with good segregation between the pairs of patient, mother and unrelated control iNCC lines.

Differential gene expression analysis revealed that there were significant differences in the expression of neural border and neural crest marker genes between patient and pooled controls. *AXIN2* expression was significantly lower (p = 0.0004) in patient iNCCs compared to pooled control iNCCs, suggesting that WNT signalling was not activated to the same extent in the patient cells by the neural crest induction medium (**Fig 4C**). In line with the reduced WNT activation in patient iNCCs, the expression of the neural border markers *PAX3* and *PAX7* was significantly lower (p = 5.9x10$^{-6}$ and p = 0.003 respectively) in patient iNCCs, while the expression of the neural crest markers *FOXD3*, *SOX10* and *NR2F1* was also significantly reduced (p = 1.37x10$^{-5}$, p = 1.14x10$^{-7}$ and p = 0.01, respectively) in patient iNCCs compared to pooled control iNCCs (**Fig 4C**). Interestingly, the expression of the neural border markers *TFAP2A*, *ZIC1* and *MSX1* was unchanged (p > 0.05) between patient iNCCs and pooled control iNCCs, while the expression of *MSX2* was significantly higher (p = 7.88x10$^{-7}$) in patient iNCCs (**Fig 4C**). Therefore, while expression of some of the earlier markers of NCC differentiation (the neural border genes) indicated that the patient cells had begun to differentiate into iNCCs, three out of four of the later neural crest identity markers examined here had significantly lower expression, indicating that the patient cells had not differentiated to the same extent as the other lines by 120h.

## Global changes in gene expression in BMKS patient iNCCs

The RNA-Seq data was examined to investigate the global differences in gene expression in patient iNCCs in relation to the pooled mother and unrelated control iNCCs. In total, DESeq2 identified 5746 DEGs (p < 0.05); 2962 genes were downregulated in patient iNCCs while 2784 genes were upregulated in patient iNCCs (**Fig 5A**, **S1 File**). The number of DEGs between the

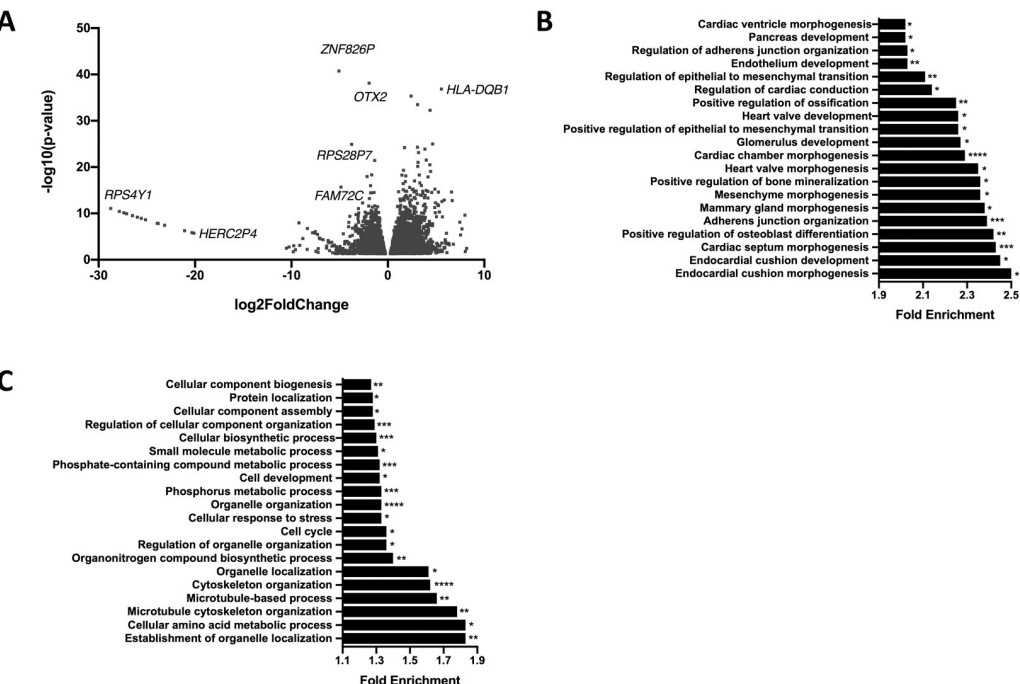

**Fig 5. RNA-Seq analysis of iNCCs.** A) Volcano plot showing all 5746 significantly differentially expressed genes (DEGs) in patient iNCCs compared to pooled mother and unrelated control (pooled control) iNCCs obtained using DESeq2 analysis of RNA-Seq data. Log2foldchange < 0 = significantly upregulated DEGs in patient iNCCs compared to pooled control iNCCs, log2foldchange > 0 = significantly downregulated DEGs in patient iNCCs compared to pooled control iNCCs. B) Top 20 enriched Gene Ontology (GO) terms for biological processes associated with DEGs in patient iNCCs compared to pooled control iNCCs, obtained using the PANTHER Overrepresentation Test for Biological Processes. C) Top 20 enriched Gene Ontology (GO) terms for biological processes associated with all differentially spliced genes in patient iNCCs compared to pooled control iNCCs, obtained using the PANTHER Overrepresentation Test for Biological Processes. * p-value < 0.05, ** p-value < 0.01, ***p-value < 0.001, **** p-value < 0.0001.

patient and pooled control iNCCs was almost five times higher than the number of DEGs in the iPSCs, suggesting that the *TXNL4A* variants in the patient had a much greater effect on the transcriptome in differentiated cells, in line with the tissue-restricted phenotype. Similar to the iPSCs, *TXNL4A* expression was significantly lower in patient iNCCs (**S6 Fig in S2 File**). No other significant change in core splicing factor expression was observed in the patient iNCC lines but increased expression of auxiliary splicing factors *SRSF8* and *ESRP-1* with decreased expression of *SRRM2* was observed (**S3 Table in S2 File, S1 File**). 5282 DEGs were uniquely differentially expressed between patient and pooled control iNCCs but not differentially expressed in the iPSCs, while 463 DEGs were differentially expressed between patient and pooled control cells in both the iPSCs and the iNCCs, and 717 of the DEGs identified in iPSCs were not differentially expressed between patient and pooled control in the iNCCs (**S1 File**).

To investigate whether particularly subsets of genes were differentially expressed in the patient iNCCs, GO enrichment analysis was performed on all the DEGs. Interestingly, among the top 20 most enriched GO terms for biological processes were positive regulation of osteoblast differentiation, positive regulation of bone mineralisation and positive regulation of ossification, regulation of the EMT, and regulation of adherens junction organisation (**Fig 5B**). This GO enrichment suggested that subsets of genes involved in bone development (important in the formation of the craniofacial skeleton), the EMT (a critical part of iNCC differentiation to become migratory) and cell-cell adhesion (important in neural crest differentiation and function, and essential for the EMT), were over-represented in the DEGs, all of which could

strongly link to defects in NCC biology and normal craniofacial development. IPA analysis of the DEGs indicated that WNT/β-catenin signalling was the second most enriched canonical pathway associated with the DEGs, while CTNNB1 (β-catenin, one of the key mediators of WNT signalling), SOX2 and WNT3A were the top three most enriched upstream regulators (**S2A, S2B Fig in** S2 File). Together, these data suggested a strong link between the DEGs and the WNT pathway, and indicated that an inefficient response to exogenous WNT signalling during the differentiation protocol in the patient cells was resulting in defective iNCC differentiation compared to the maternal and unrelated control cells.

The top molecular and cellular function identified by IPA as most altered in the patient-derived cells was cellular movement (**S2D Fig in** S2 File). Given that cellular movement and migration requires the differentiating iPSCs to undergo an EMT and the GO analysis of the DEGs identified regulation of the EMT as an enriched process associated with the DEGs, these data suggested that there might be differences in cell migration between patient, maternal and unrelated control lines resulting from alterations in the EMT. IPA also identified enriched functional networks relating to embryonic development and developmental disorders (**S2C Fig in** S2 File).

## Mis-splicing in BMKS patient iNCCs

rMATS was used to identify and quantify differential splicing events between the patient and pooled controls in the iNCC RNA-Seq data. In total, rMATS identified 2991 differential splicing events in 2029 different genes (p < 0.05, FDR < 0.05), 589 of which were also differentially expressed (**Fig 6A**, **S1 File**). Several of the altered splicing events were validated by RT-PCR (**S7 Fig in** S2 File).

As with the iPSCs, by far the most prevalent form of differential splicing between the patient and pooled control iNCCs was exon skipping (SE) (**Fig 6A**). Multiple different forms of mis-splicing were observed in 239 genes, including one gene (*COMMD4*) which presented with four different forms of differential splicing between the patient and pooled control iNCC lines but did not show any forms of mis-splicing in the iPSCs (**Fig 6B**, **S6 and S7 Tables in** S2 File).

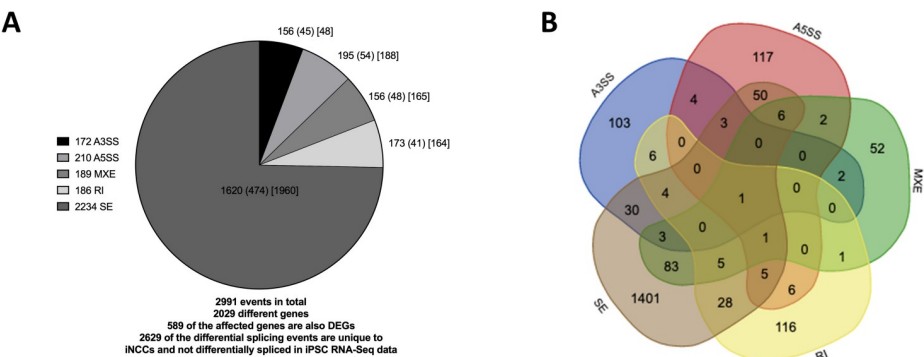

**Fig 6. Differential alternative splicing events in iNCCs.** A) Number of significantly different alternative 3' splice site (A3SS) and alternative 5' splice site (A5SS) events, mutually exclusive exon (MXE) events, intron retention events (RI) events and exon skipping (SE) events in pooled patient iNCCs compared to pooled maternal and unrelated control iNCCs, as determined by rMATS. The total number of events for each class of differential splicing is shown in the legend, the number of affected genes for each class of event shown in the corresponding segments of the pie chart, the number of the genes which are also differentially expressed for each class of event shown in round brackets, and the number of differential splicing events for each class unique to iNCCs and not observed as a differential splicing event between patient iPSCs and pooled control iPSCs in the iPSC RNA-Seq data is shown in square brackets. B) Venn diagram showing the numbers of differentially spliced genes in patient iNCCs compared to pooled control iNCCs obtained by rMATs.

Interestingly, mis-splicing was observed for the core splicing factor *SNRPA* and auxiliary splicing factors *SRRM2*, *ESRP-2*, *RBFOX2*, *and MBNL1* in the patient iNCC lines (**S3 Table in S2 File**). As ESRP-2, RBFOX2 and MBNL1 are involved in regulating genes involved in the EMT their mis-splicing may be one of the contributing factors to the EMT defect of patient iNCCs [71–73].

To identify whether particular subsets of genes were mis-spliced, all 2029 differentially spliced genes between the patient and pooled controls in the iNCCs were analysed by GO enrichment analysis. Among the most enriched GO terms were terms associated with organelle and cytoskeletal organisation, the cell cycle and cellular response to stress (**Fig 5C**). Among the top canonical pathways identified by IPA analysis of the mis-spliced genes was epithelial adherens junction signalling (**S8A Fig in S2 File**). Given the importance of adherens in EMT, this mis-splicing could contribute further to the potential EMT and cell migration defects in patient iNCCs identified from the analysis of the DEGs [74]. While TP53 was one of the most enriched upstream regulators of the mis-spliced genes, we did not identify any differences in apoptosis levels suggesting that any alterations in splicing of TP53 target genes were not having a functional impact on cell death (**Fig 4A**, **S8B Fig in S2 File**). Among the top associated networks with the mis-spliced genes were embryonic developmental and developmental disorders, relevant to the BMKS phenotype (**S8C Fig in S2 File**). IPA analysis was also performed with the mis-spliced genes observed only in iNCCs and not in iPSCs, and also with the genes showing significantly differential SE events between patient and pooled controls (the only subset of mis-spliced genes with sufficient events to perform reliable enrichment analysis) (**S8 Fig in S2 File**). However, because the majority of the mis-splicing events were unique to the iNCCs and were SE events, the upstream regulators, associated canonical pathways and associated networks identified by these analyses overlapped closely with those from the analysis of all the mis-spliced genes.

One interesting observation from this splicing analysis was the finding that ESRP-1 was among the top 10 most enriched upstream regulators in the iNCC-specific mis-spliced genes (**S8B Fig in S2 File**). ESRP-1 is an epithelial cell type-specific splicing regulator and regulates the splicing of a number of transcripts which undergo alterations in splicing during the EMT, providing further evidence for potential aberrations in the EMT in patient iNCCs [75–77]. Interestingly, *ESRP-1* expression was significantly higher in patient iNCCs compared to pooled control iNCCs (**S1 File**). *ESRP-1* expression decreases during the EMT as cells transition from an epithelial state to a mesenchymal state, suggesting that patient iNCCs had remained in a more epithelial state by 120h differentiation [78,79]. The significantly higher *ESRP-1* expression in patient iNCCs was in agreement with the findings that patient iNCC cells had not differentiated to the same extent as the mother and unrelated control iNCC cells by 120h.

## Defects in the Epithelial-to-Mesenchymal Transition (EMT) in BMKS patient iNCCs

Given the importance of the EMT in NCC specification and the absolute requirement for NCCs to migrate *in vivo*, any delays or defects in the EMT in NCCs would be expected to have an impact on development. We therefore investigated the EMT in more detail in the patient iNCCs. ESRP-1 regulates the splicing of many genes, including regulating the mutually exclusive inclusion of *FGFR2* exon 8, of which there are two isoforms, one of which is found in epithelial *FGFR2* transcripts (e.g. RefSeq NM_022970) and the other found in mesenchymal *FGFR2* transcripts (e.g. RefSeq NM_000141) [75–77,80]. ESRP-1 promotes inclusion of the epithelial exon 8, and as ESRP-1 expression decreases during the EMT there is a switch to the inclusion of the mesenchymal exon 8. Comparing patient iNCCs to pooled control iNCCs,

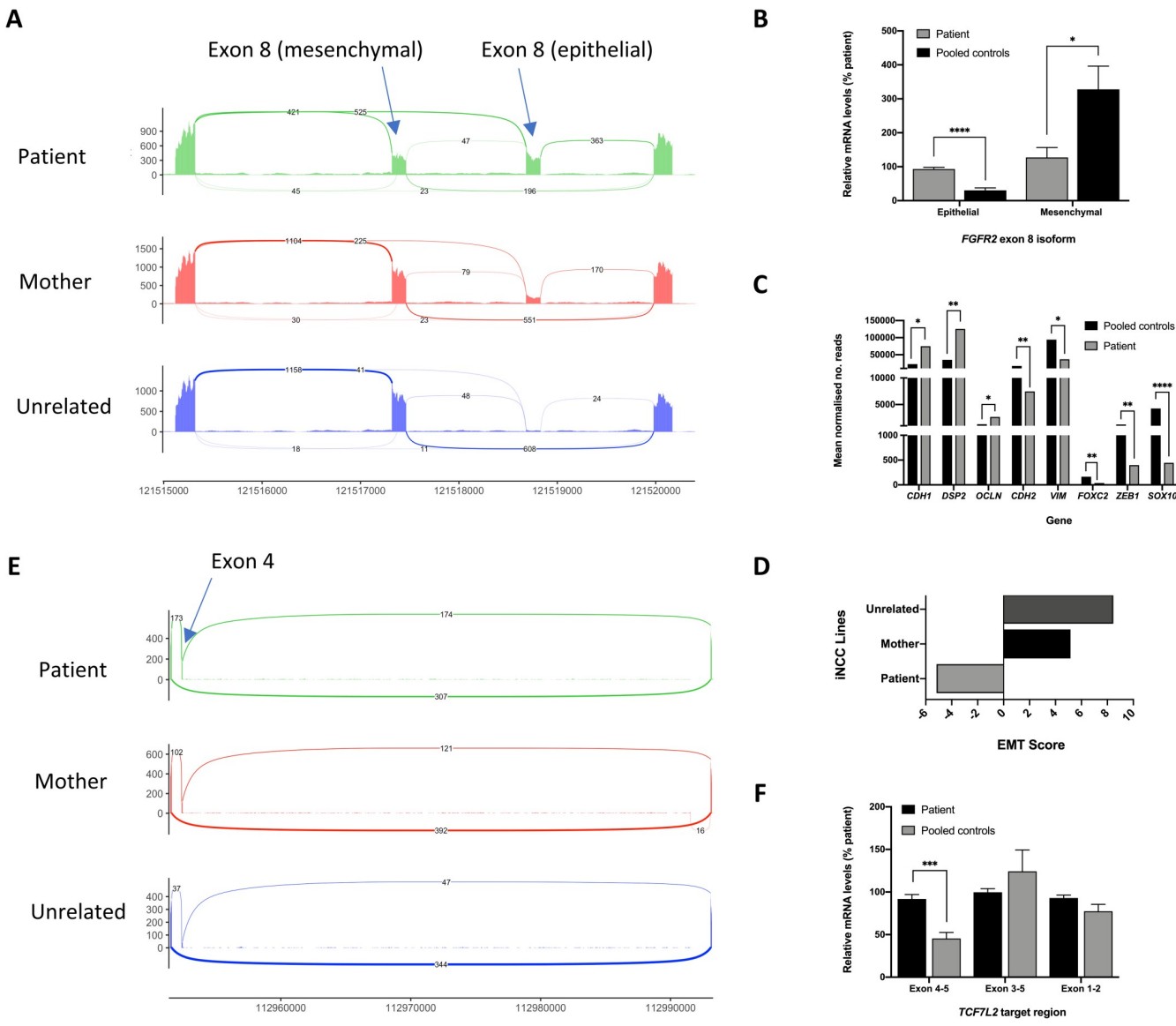

**Fig 7. Defects in the EMT in BMKS patient iNCCs.** A) Sashimi plot showing altered inclusion of the epithelial and mesenchymal isoforms of *FGFR2* exon 8 in pooled patient, pooled mother and pooled unrelated control iNCCs, derived from rMATS analysis of RNA-Seq data. B) Relative *FGFR2* epithelial exon 8 and mesenchymal exon 8 mRNA expression levels for pooled patient iNCCs compared to pooled parent and unrelated control (pooled control) iNCCs, determined using qPCR of cDNA from each cell line. Graphs were obtained using the $\Delta\Delta C_T$ method with *ACTB* as the endogenous reference gene and normalised to the KW181A patient line. n = 3. C) Key marker gene expression levels in iNCCs from RNA-Seq data. Mean normalised read counts for epithelial (*CDH1*, *DSP2*, *OCLN*) and mesenchymal (*CDH2*, *VIM*, *FOXC2*, *ZEB1*, *SOX10*) marker genes. D) EMT scores for pooled patient, pooled mother and pooled unrelated iNCCs calculated from RNA-Seq data using the method described by Chae et al., (2018). E) Sashimi plot showing differential splicing of *TCF7L2* exon 4 from RNA-Seq data, determined by rMATS analysis. F) Relative *TCF7L2* exon 4 inclusion (exon 4–5) compared to overall *TCF7L2* expression (exon 3–5 and exon 1–2) in patient iNCCs compared to pooled mother and unrelated control (pooled control) iNCCs, determined using qPCR of cDNA from each cell line. Graphs were obtained using the $\Delta\Delta C_T$ method with *ACTB* as the endogenous reference gene and normalised to the KW181A patient line. n = 3. * p-value < 0.05, ** p-value < 0.01, *** p-value < 0.001, **** p-value < 0.0001.

there was significantly higher expression of the epithelial *FGFR2* exon 8 and significantly lower expression of the mesenchymal *FGFR2* exon 8, which was confirmed by qPCR (**Fig 7A and 7B**). Additionally, ESRP-1 regulates the splicing of other genes which undergo EMT-specific changes in splicing, including *CTNND1*, *CD44* and *ENAH* [81]. Significant differences in *CTNND1* and *CD44* splicing between patient and pooled control iNCCs were identified by

rMATs (**S1 File**). Furthermore, expression of other epithelial marker genes, such as *CDH1*, *DSP2* and *OCLN* was significantly higher in patient iNCCs, while expression of several mesenchymal marker genes, including *CDH2*, *VIM*, *FOXC2*, *SOX10* and *ZEB1*, were all significantly higher in pooled control iNCCs (**Fig 7C**). Using the method described by Chae et al., (2018), the EMT score for the patient, mother and unrelated control lines was calculated from the RNA-Seq data as an indication of how 'epithelial' (low EMT score) or 'mesenchymal' (high EMT score) the cell lines were [82]. The patient iNCCs had a score of -5, while the mother iNCCs had an EMT score of +5 and the unrelated controls had an EMT score of +8 (**Fig 7D**). Furthermore, extending the iNCC differentiation protocol time to 168h in the patient iNCCs did not allow the patient iNCCs to 'catch up' with the parent and unrelated control cells (**S9 Fig in S2 File**).

## Mis-splicing of *TCF7L2* in BMKS patient iNCCs

The RNA-Seq data clearly indicated that patient iNCCs had not responded to WNT signalling to the same extent as mother and unrelated control iNCCs during differentiation, leading to diminished differentiation including a severely delayed or defective EMT. The question remained as to why patient cells did not respond to WNT signalling fully and how this lack of response might relate to reduced *TXNL4A* expression levels. We postulated that a critical gene early in the WNT signalling pathway was directly mis-spliced by the reduced *TXNL4A* expression, and this mis-splicing event compromised the WNT response. One of the difficulties in interpreting the differential gene expression and differential splicing in the RNA-Seq data is determining direct effects (i.e. effects stemming from the change in *TXNL4A* expression itself) from indirect effects (i.e. effects arising because a regulatory gene upstream of the gene in question was mis-spliced or mis-expressed). Presumably mis-splicing of a few critical genes would be sufficient to lead to a cascade of altered expression and splicing of the affected genes, which may affect genes in the critical signalling pathways of differentiation, in this case WNT signalling. Our iNCC induction protocol activated WNT signalling by inhibiting GSK3; therefore, any defects in the WNT pathway here would have to affect WNT components downstream of GSK3 [32]. It should be noted that in patients *in vivo*, WNT signalling is activated endogenously, so mis-splicing of components upstream of GSK3 could also affect the WNT response.

We therefore investigated mis-splicing of core components of the WNT pathway downstream of GSK3 in our iNCC RNA-Seq data. A significant SE event was identified in *TCF7L2* with increased inclusion of *TCF7L2* exon 4 in the patient iNCCs, although there was no change in overall *TCF7L2* gene expression between patient and pooled control iNCCs (**Fig 7E**). This altered inclusion of *TCF7L2* exon 4 without an increase in overall gene expression was verified by qPCR (**Fig 7F**) and was not observed in the patient iPSC rMATS data, suggesting an iNCC-specific differential splicing event.

*TCF7L2* encodes TCF4, one of four TCF factors in vertebrates which interacts with β-catenin in the nucleus and then binds to the promoters of WNT target genes to modulate their expression and trigger the WNT signalling transcriptional response [83,84]. Both human and mouse *TCF7L2* contain 17 exons, with exon 4, exon 8 and exons 13–16 being alternatively spliced in various combinations. Mouse *Tcf7l2* transcripts containing exon 4 lead to a dampened WNT transcriptional response compared to transcripts excluding exon 4 [83]. The findings that patient iNCCs displayed a defective WNT response and patient iNCCs had increased exon 4-containing *TCF7L2* transcripts compared to pooled control iNCCs could explain, at least in part, the failure to fully respond to exogenous WNT pathway activation and the failure of patient iNCCs to differentiate to the same extent as parent and unrelated control lines.

## Features of skipped exons in BMKS patient iNCCs

Finally, we turned our attention to why particular subsets of genes were mis-spliced in patient iNCCs. In particular, we focused on the sequence properties of exons that are significantly more or less skipped in patient iNCCs compared to pooled controls in the RNA-Seq data. Each set of exons was compared to two different control sets of exons. Control sequences were comprised of a central exon, plus its two flanking introns and two flanking exons (E-I-E-I-E) derived from genes containing the SE event but for an exon not significantly differentially skipped (internal control exons) and from exons from highly expressed genes which were not differentially expressed between patient and pooled control cells in the iNCC RNA-Seq data which were not differentially spliced (external control genes). These extended control sequences were used because splicing is influenced by numerous signals, both within the intron to be removed and within the surrounding exonic and intronic sequences.

For exons more or less skipped in patient iNCCs than in pooled control iNCCs we investigated skipped exon length and lengths of upstream and downstream introns and exons; GC content of the skipped exon and of the upstream and downstream introns; upstream and downstream intron branch point to 3' splice site (BPS-3'SS) distance; strengths of splice donor and splice acceptor sequences of the SE; and strength of upstream splice donor and downstream splice acceptor site (**Table 2**, **S10 Fig in S2 File**). We found that exons more skipped in patient iNCCs were shorter than both internal (p < 0.0001) and external (p < 0.0001) control

**Table 2. Properties of significantly differentially skipped exons in iNCCs.**

|  | Exons more skipped in patient | | Exons less skipped in patient | |
|---|---|---|---|---|
|  | Compared to internal control exons | Compared to external control exons | Compared to internal control exons | Compared to external control exons |
| Skipped exon length | Shorter (****) | Shorter (****) | Shorter (*) | NS |
| Upstream exon length | Shorter (**) | NS | NS | Longer (*) |
| Downstream exon length | NS | Shorter (**) | NS | Shorter (*) |
| Upstream intron length | Longer (***) | Longer (****) | Longer (****) | Longer (****) |
| Downstream intron length | Longer (****) | Longer (****) | Longer (****) | Longer (****) |
| Skipped exon GC content | NS | Higher (****) | NS | NS |
| Upstream intron GC content | NS | NS | NS | Higher (***) |
| Downstream intron GC content | NS | NS | NS | Higher (****) |
| Upstream intron BPS-3'SS distance | Longer (****) | Longer (****) | Longer (**) | Longer (**) |
| Downstream intron BPS-3'SS distance | Longer (*) | Longer (**) | Longer (****) | Longer (****) |
| Skipped exon splice donor strength | Weaker (****) | Weaker (****) | Weaker (****) | Weaker (****) |
| Skipped exon splice acceptor strength | Weaker (****) | Weaker (****) | Weaker (****) | Weaker (****) |
| Upstream splice donor strength | NS | NS | NS | NS |
| Downstream splice acceptor strength | NS | NS | NS | NS |

Summary of the properties of exons more or less skipped in pooled patient iNCCs compared to pooled mother and unrelated control iNCCs. Asterisks indicate statistical significance.

* p-value < 0.05

** p-value < 0.01

*** p-value < 0.001

**** p-value < 0.0001.

exons, had shorter upstream exons than internal control exons (p = 0.005) and shorter downstream exons than external control exons (p < 0.0001) (**Table 2**, **S10 Fig in S2 File**). The exons more skipped in patient iNCCs were also associated with significantly longer upstream introns than both internal (p = 0.0006) and external (p = 0.0041) control exons, significantly longer downstream introns than both internal (p < 0.0001) and external (p < 0.0001) control exons, had a significantly higher GC content than external control exons (p < 0.0001), had significantly longer upstream and downstream intron BPS-3'SS distances than both internal (p < 0.0001 for upstream, p = 0.0117 for downstream) and external (p < 0.0001 for upstream, p = 0.0022 for downstream) control exons, and were associated with weaker splice donor sites than internal and external controls (p < 0.0001 for both sets of controls) and significantly weaker splice acceptor sites than internal and external controls (p < 0.0001 for both sets of controls) (**Table 2**, **S10 Fig in S2 File**).

For the exons less skipped in patient iNCCs, the skipped exons were shorter than internal control exons (p = 0.0366), associated with significantly longer up- and downstream introns than both internal (p < 0.0001 for both upstream and downstream introns) and external (p < 0.0001 for upstream introns, p = 0.0268 for downstream introns) control exons, were associated with up- and downstream introns with significantly higher GC content than external control exons (p = 0.0004 for upstream introns, p < 0.0001 for downstream introns), and had longer upstream and downstream BPS-3'SS distances than both internal (p = 0.0015 for upstream, p < 0.0001 for downstream) and external (p = 0.0064 for upstream, p < 0.0001 for downstream) control exons. The exons also had weaker splice donor and splice acceptor sites than both internal (p < 0.0001 for both donors and acceptors) and external (p < 0.0001 for donors and acceptors) control exons (**Table 2**, **S10 Fig in S2 File**). These findings suggest that *cis*-features of exons make certain exons more or less vulnerable to skipping events in iNCCs when the expression of *TXNL4A* is reduced.

## Discussion

How variants in core pre-mRNA splicing factors, required for splicing of all pre-mRNAs, lead to tissue-specific developmental defects such as the craniofacial disorder BMKS remains largely unknown. Here, we generated the first human cell model of BMKS using iPSCs and differentiated these iPSCs into iNCCs as a specific and disease-relevant system to investigate BMKS *in vitro*. We found that iPSCs from the BMKS patient proliferated significantly slower with RNA-Seq analysis revealing considerable differences in gene expression and pre-mRNA splicing. Furthermore, failure of patient iPSCs to differentiate to iNCCs fully and undergo an EMT provides a direct link to potential defects in craniofacial development in the patient, either through a simple delay in differentiation or a failure of NCCs to ever reach a fully differentiated state. The defective differentiation of patient iPSC cells results from the observed diminished WNT signalling response likely arising from mis-splicing of critical genes in the WNT pathway such as *TCF7L2*. Additionally, the conserved functional and physical properties of the pre-mRNAs that are mis-spliced in patient iNCCs suggests that particular exons are more vulnerable to mis-splicing when *TXNL4A* expression is reduced, although the mechanism behind this vulnerability remains unclear.

While this study investigated a single BMKS family, and the results are therefore not necessarily specific to all BMKS patients, there is good evidence for major differences in this particular patient's iNCCs compared to those of her unaffected mother and several unrelated individuals. Furthermore, by pooling the RNA-Seq data from the mother's lines with data from unrelated control lines, we considered differential expression or mis-splicing events as significant only if different between patient cells and all other individuals in this study, thereby

reducing genetic background effects on the results. Nonetheless, preliminary data from the single BMKS family here provides impetus and a clear argument for extending iPSC generation and differentiation to further BMKS families to determine which, and to what extent, differences identified in this BMKS patient are common between all BMKS patients with different genotypes. Furthermore, to rule out effects of the differentiation protocol itself on differences in iNCC properties between individuals, it would be worthwhile repeating the iPSC-to-iNCC differentiation using an alternative protocol to that used here [32].

Our work has revealed a number of differences between the patient iNCCs and iNCCs from her mother and the unrelated individuals which can be linked to aberrant craniofacial development, in particular the finding that patient cells do not differentiate to the same extent and do not fully undergo an EMT. Indeed, there are reports of other disorders with craniofacial phenotypes whereby patient NCCs do not differentiate as well or show delayed NCC differentiation compared to those of unaffected individuals [31,33,39]. Our data cannot discriminate between several possibilities for delayed NCC differentiation and EMT which could link to craniofacial defects *in vivo*:

1. Patient cells simply differentiate and undergo an EMT slower and would, eventually, 'catch up'.

2. Patient cells never fully differentiate.

3. Only a portion of patient cells differentiated to iNCCs (stochastic differentiation).

Of particular interest is impairment of EMT in patient cells; while likely to derive from delayed differentiation, undergoing an EMT is absolutely essential during neural crest induction as migratory capacity of NCCs is vital for vertebrate development. Defects in NCC migration have been observed in differentiated iNCCs derived from Treacher Collins syndrome (TCS) patients and in mouse models of TCS [36,37,85]. Defective NCC migration in Bardet-Biedl syndrome is thought to underlie the craniofacial phenotype, demonstrating that defective NCC migration can be pathogenic [33,86]. Investigating migratory capacity of BMKS patient iNCCs compared to non-patient iNCCs will be an interesting further line of characterisation.

As might be expected, we observed considerably more differences between cell lines at the differentiated iNCC stage than at the pluripotent stage. There is less clustering between mother and patient lines in the RNA-Seq data and more differentially expressed and mis-spliced genes, suggesting that effects of *TXNL4A* variants in patients are magnified upon differentiation to iNCCs. This finding is in agreement with previous work which has shown greater disparity between disease-affected and disease-unaffected differentiated cells than iPSCs [87,88]. This observation would also go some way in explaining the tissue-specificity of BMKS; if NCCs are particularly sensitive to *TXNL4A* expression changes and show the greatest cell type functional and transcriptomic differences, then tissues derived from NCCs would be phenotypically most affected. The large number of mis-expressed and mis-spliced genes in both patient iPSCs and iNCCs is unsurprising given that *TXNL4A* is a spliceosome factor required for the splicing of all pre-mRNAs, so changes in *TXNL4A* expression should affect a large number of genes. Interestingly, other studies investigating reduction in expression or inhibition of other core spliceosome factors have found global changes in the expression of other spliceosome factors, suggesting an integrated network of splicing factor gene expression [58,89,90]. However, we did not observe any global changes in core or auxiliary splicing factor expression in our BMKS iNCCs. Why altering the expression of only some core splicing genes is linked to global changes in spliceosome gene expression remains to be seen, although the extent of the reduction of expression or specific function of the core splicing factor may play a role.

The findings from patient iNCC RNA-Seq data that WNT signalling is not activated properly during differentiation could explain why NCCs are more sensitive to changes in *TXNL4A* expression. Furthermore, the identification of splicing defects in key genes in the WNT pathway, in particular increased *TCF7L2* exon 4 inclusion in patient iNCCs, linked with dampened WNT target gene activation, help to explain the WNT signalling defects [83]. However, WNT signalling is involved in the differentiation of a large number of other cell types, including cells of endothelial, cardiac, vascular smooth muscle and haematopoietic lineages, and so defects in other tissues might be expected if the WNT response is defective [91–94]. While extra-craniofacial abnormalities are seen in some BMKS patients, including congenital heart defects, the facial region is by far the most affected system [4,6]. The extent to which WNT signalling plays a role in the specification of different tissues, the importance of individual WNT components such as *TCF7L2* in different tissues, and the extent of mis-splicing of factors such as *TCF7L2* in different cell and tissue types, may not be uniform. Therefore, it may be that defects in WNT signalling primarily affecting NCC specification also affect the specification of other tissues but to a lesser extent, and the degree to which these other tissues are affected by the WNT defects determines whether the system in question shows a phenotype. However, this hypothesis requires further experimental evidence to unravel.

In line with a number of other reports of mis-splicing in human spliceosome factor knockdown or inhibition models, we found SE by far the most prevalent form of mis-splicing in both the BMKS patient-derived iPSCs and iNCCs [58,89]. Whether this prevalence of SE in human cells is because human transcripts are more susceptible to SE events than other forms of mis-splicing, or because SE events are easier to detect bioinformatically, is unclear. Analysis of the physical properties of SEs in our BMKS iNCCs has characterised exon features particularly sensitive to reduced *TXNL4A* expression. Exons skipped both more and less in patient iNCCs shared features such as significantly longer proximal introns, longer proximal intron BPS-3'SS distances and weaker splice acceptor and splice donor strengths. Links between increasing upstream BPS-3'SS distance and SE events in mammalian cells, and between increased lengths of flanking introns and SE, have been reported previously [56,58,95]. While it is unclear precisely why such features render exons more vulnerable to skipping, we postulate that short exons surrounded by long introns with long BPS-3'SS distances and weak splice sites may be more difficult for the spliceosome to locate and assemble around. Reducing the expression of a core spliceosome factor such as *TXNL4A* would affect spliceosome composition or function and then 'tip the exons over the edge' and lead to skipping.

BMKS is just one of five human craniofacial developmental disorders caused by variants in core spliceosome or exon junction complex (EJC) components [6,17]. While there is overlap between phenotypes of patients with these disorders, and genetic and physical connections have been established between the causative spliceosomal factors, the precise craniofacial and extra-craniofacial features of each disorder are unique. Therefore, while commonalities in underlying disease mechanisms might be expected, it is likely that there are also differences in the aetiologies. For example, iPSCs from Richieri-Costa-Pereira syndrome (RCPS) patients with variants in the EJC component *EIF4A3* differentiated to iNCCs as well as control iPSCs, the patient iNCCs showed no defects in proliferation or apoptosis, but had defects in migration and in differentiation to NCC-derived mesenchymal stem-like cells (nMSCs) [24]. These findings *in vitro* were supported by the specific defects observed in the craniofacial skeleton in *Eif4a3* haploinsufficient mouse models of RCPS [24]. *Sf3b4*-knockdown Xenopus embryos, modelling haploinsufficiency of the U2 snRNP component *SF3B4* in Nager syndrome (NS), revealed reduced expression of a number of NCC marker genes at the neural plate border associated with neural plate broadening. Furthermore, there were reduced numbers of NCCs and hypoplasia of NCC-derived craniofacial cartilages later in development in *Sf3b4*-deficient

tadpoles [39]. There are certainly parallels between these observations in NS models and our observations in BMKS patient-derived iNCCs. Finally, there have been *in vitro* and *in vivo* models of Mandibulofacial Dysostosis Guion-Almeida type (MFDGA), caused by haploinsufficiency of *EFTUD2*, the craniofacial disorder with the strongest phenotypic overlap with BMKS and the strongest genetic link as both disorders are caused by U5 snRNP gene variants [58–60,96,97]. In zebrafish, *eftud2* is broadly expressed but enriched in the developing craniofacial region, while in the mouse *Eftud2* is also widely expressed with an enrichment in the head, brachial arches and developing brain during early development [17,59,96–98]. Human *EFTUD2*-knockdown HEK293 cells revealed abnormal proliferation similar to our BMKS patient iPSCs [58]. A combination of these *EFTUD2* knockdown human cells and several zebrafish models of MFDGA have implicated aberrant endoplasmic reticulum stress and activation of p53-dependent apoptosis as a key disease mechanism [17,58–60]. In contrast, we observed no differences in apoptosis between BMKS patient, mother and unrelated control iPSCs or iNCCs suggesting disease processes of MFDGA and BMKS are certainly not identical. Furthermore, comparison of GO and IPA enrichment analysis of the differentially expressed and mis-spliced genes in BMKS iNCCs and the *EFTUD2* knockdown HEK293 cells displayed enrichment of different groups of genes with different associated regulators and pathways in each disease model [58]. It is possible that these differences arise from differences in cell type, but it seems likely reducing expression of *EFTUD2* does not affect identical groups of genes as reducing expression of *TXNL4A*, at least in part explaining the phenotypic differences between the two disorders. In the future, it would be interesting to investigate the effects of reduced *TXNL4A* expression *in vivo* in mouse and/or zebrafish models, as has been done for other spliceosomal craniofacial disorders.

Taken together, this work has established novel human BMKS model iPSCs and their differentiation into induced neural crest cells. The initial cellular, transcriptomic and splicing characterisation presented here has provided insight into how reducing *TXNL4A* expression might impair the differentiation and functions of NCCs leading to craniofacial defects. Further characterisation of the patient-derived iNCCs, in particular investigating cell migration, and confirming the findings presented here in additional BMKS families, should cast further light on how reduced *TXNL4A* expression specifically leads to craniofacial defects and any potential therapeutics targeting the associated molecular defects.

## Supporting information

**S1 Data.**
(XLSX)

**S1 File.**
(XLSX)

**S2 File.**
(PDF)

## Acknowledgments

The authors would like to thank Dr William Barrell (King's College London) for his advice and experience with the neural crest cell differentiation protocol. The authors also thank members of the Genomic Technologies Facility (University of Manchester) and Dr Leo Zeef of the Bioinformatics Facility (University of Manchester) for their assistance with the RNA-Seq experiments and analysis; Alexander Moore for his assistance with the analysis of RNA-Seq data; Professor Gareth Howell of the Flow Cytometry Facility (University of Manchester) for

his technical assistance with flow cytometry experiments; Dr Steve Marsden of the Bioimaging Facility (University of Manchester) for his assistance with microscopy experiments; and members of the O'Keefe, Hentges, Newman and Kimber laboratories for helpful discussions and technical assistance.

## Author Contributions

**Conceptualization:** Katherine A. Wood, Sofia Douzgou, Susan J. Kimber, William G. Newman, Raymond T. O'Keefe.

**Data curation:** Katherine A. Wood, Charlie F. Rowlands, Huw B. Thomas, Raymond T. O'Keefe.

**Formal analysis:** Katherine A. Wood, Charlie F. Rowlands, Huw B. Thomas, Steven Woods, Sofia Douzgou, Raymond T. O'Keefe.

**Funding acquisition:** Susan J. Kimber, William G. Newman, Raymond T. O'Keefe.

**Investigation:** Katherine A. Wood, Charlie F. Rowlands, Huw B. Thomas, Steven Woods, Julieta O'Flaherty, Sofia Douzgou.

**Methodology:** Steven Woods, Julieta O'Flaherty, Susan J. Kimber.

**Project administration:** William G. Newman, Raymond T. O'Keefe.

**Resources:** Steven Woods, Julieta O'Flaherty, Sofia Douzgou, Susan J. Kimber, William G. Newman.

**Software:** Charlie F. Rowlands.

**Supervision:** Susan J. Kimber, William G. Newman, Raymond T. O'Keefe.

**Validation:** Katherine A. Wood, Huw B. Thomas.

**Visualization:** Katherine A. Wood.

**Writing – original draft:** Katherine A. Wood, William G. Newman, Raymond T. O'Keefe.

**Writing – review & editing:** Katherine A. Wood, Charlie F. Rowlands, Huw B. Thomas, Steven Woods, Julieta O'Flaherty, Sofia Douzgou, Susan J. Kimber, William G. Newman, Raymond T. O'Keefe.

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
