## [Decision Letter · Decision Letter 0]

21 May 2020

PONE-D-20-13089

Modelling the developmental spliceosomal craniofacial disorder Burn-McKeown syndrome using induced pluripotent stem cells

PLOS ONE

Dear Raymond,

Thank you for submitting your manuscript to PLOS ONE. As you will see, both reviewers were generally very positive and they both agreed that the manuscript would be suitable for publication after Minor Revisions. Therefore, we invite you to submit a revised version of the manuscript that addresses the points raised during the review process.

We look forward to receiving your revised manuscript.

Kind regards,

Emanuele Buratti, Ph.D.

Academic Editor

PLOS ONE

Reviewers' comments:

Reviewer's Responses to Questions

**Comments to the Author**

1. Is the manuscript technically sound, and do the data support the conclusions?

Reviewer #1: Yes

Reviewer #2: Yes

2. Has the statistical analysis been performed appropriately and rigorously? 

Reviewer #1: Yes

Reviewer #2: Yes

3. Have the authors made all data underlying the findings in their manuscript fully available?

Reviewer #1: Yes

Reviewer #2: Yes

4. Is the manuscript presented in an intelligible fashion and written in standard English?

Reviewer #1: Yes

Reviewer #2: Yes

5. Review Comments to the Author

Reviewer #1: In this manucript, carefully prepared by Wood et al., the authors describe their experimental approach to establishing the link between congenital mutations of TXNL4A (a compoent of core spliceosomal subunit U5 snRNP) and the predominant craniofacial phenotype of the stemming Burn‑McKeown syndrome. The team has prepared induced pluripotent stem cells (iPSCs) of the patient's, patient mother's and unrelated controls' PBMCs and induced its differentiation into neural crest cells (iNCCs). Upon this induction, the authors observed slower proliferation and partly defective differentiation of the patient's iPSCs compared to mother and controls, but not any difference in apoptosis. In both cell types (iPSCs and iNCCs) the authors obseved differential expression and splicing when comparing patient and controls (including mother), both of which were more pronounced in the iNCCs. The differentially expressed and spliced genes were enriched in pathways and processes connected to NCC differentiation and function, e.g. the EMT, which is a substantial part of the NCC differentiation. Finally, the authors characterized the subset of exons that were differentially skipped in the patient iNCCs compared to controls and observed some conserved properties among these exons in terms of their length, length of theis surrounding introns, etc.

The manuscript is clearly and rigorously written. The methods and experiements are described into sufficient detail, the discussion is a little bit longish but comprehensible and logically evolving. The experiments seem to be rigorously contucted and analysed, well corroborating the findings. I have only minor comments that can be found below.

Minor comments:

page 18: „Unlike all other patients with BMKS described to date, the patient displays developmental delay and severe intellectual disability; however, whole exome sequencing has not revealed any genetic variants that could account for this significant cognitive impairment.“

Was the patient tested for structural variants? In any case, the possibility of a structural variant underlying the cognitive imapairment ought to be mentioned.

The text is sometimes a little bit longish. There are a few passages that describe numbers that can be easily observed from a table or a figure, e.g. at page 31, lines 681-684.

Lines 922-923: „However, we did not observe any global changes in splicing factor expression in our BMKS iNCCs.“ As I understand it, the authors did not observe any significant changes in the level of splicing factors expression (except of TXNL4A). However, potentially even splicing changes between patient and controls in a splicing regulator that could be NCC-specific or just important for EMT might be of high relevance, possibly explaining the differences in mis-splicing between iPSCs and iNCC, at least in part. Did the authors observe the results in that way?

Line 566: one of the control iPCS lines is depicted here as „SW71A“ but as SW171A elsewhere. I guess this is a typing error.

Lines 667-669: „For these genes which were not expressed in either the iPSCs or the iNCCs, it was unclear whether, if the gene had been expressed in both iPSCs and iNCCs, it would have been differentially spliced in both cell types.“ In fact, I do not understand why you mention that. It seems very logic and clear to me, but I see not point in particular commenting that.

Line 747-748: typping error: „S6 Fig.“ (reverse words order than usually used in the manuscript).

Line 916: one of the „mis-expressed“ words should be replaced by „mis-spliced“ I guess.

Reviewer #2: Wood and colleagues study a rare craniofacial developmental disorder, the Burn-McKeown syndrome, caused by the 34 base pair deletion in the promoter region of one allele of TXNL4A gene, combined with a loss-of-function variant on the other allele, resulting in reduced TXNL4A expression. TXNL4A gene encodes a component of the U5 snRNP, one of the core components of the spliceosome. The authors hypothesize that the TXNL4A mutations observed in BMKS patients can cause reduced assembly of the human tri-snRNP, thus affecting the splicing of a specific subset of pre-mRNAs. To test this hypothesis, they generate induced pluripotent stem cell (iPSC) lines from a patient with BMKS and her unaffected carrier mother, and from unrelated, unaffected individuals, and characterize the developed models for proliferation, differentiation and gene expression. RNA-Seq analysis revealed transcriptome-wide mis-expression and mis-splicing, in particular aberrant exon skipping, affecting distinct subsets of pre-mRNAs, with an enrichment for genes involved in processes important in NCC specification, highlighting a decreased response to the WNT signaling.

Overall, the experiments are well conducted and the obtained results could be relevant for the potential advances in the rare BMKS disease. However, the manuscript is quite dispersive in the current form and too long; it would benefit of a general shortening, especially in the introduction and discussion sections. Therapeutic potential of the developed findings should be discussed.

A few clarifications and key control experiments are nonetheless required:

1. The experiments in Figure 3A need to be quantified in some way. Ideally, the authors should perform a morphological analysis by using specific markers of differentiation. Quantification of their signals should be performed.

2. The GO analyses are not convincing and could be ameliorated by changing the selected parameters, including the EASE score and the graphical representation. In particular, in figure 4A and 4B it seems that each GO term displays the same fold enrichment. P value for each GO term should be added.

3. Validation of GE changes were performed using the patient sample and the pooled control. Validations should also be performed by taking separately samples from mother and unrelated control.

Minor points:

Figure 1A – Western blot analysis to show and quantify TXNL4A protein expression in the derived iPSCs should be performed. In the figure legend please specify that the qPCR analysis is performed to monitor TXNL4A expression.

Table S3, S6, S7 and S10: It would be suitable to represent the results of the Ingenuity Pathway Analyses by using charts and cluster dendrograms or pathway graphs.

Table 2 and Table 3 could be represented with graphs or pie charts.

6. PLOS authors have the option to publish the peer review history of their article (what does this mean?). If published, this will include your full peer review and any attached files.

Reviewer #1: No

Reviewer #2: No

---

## [Author Response · Author response to Decision Letter 0]

14 Jun 2020

Emanuele Buratti, Ph.D.

Academic Editor

PLOS ONE

Dear Emanuele,

Thank you for considering out manuscript. We have now revised our manuscript according to the comments of the two reviewers. Please find below a point by point response to their comments and a description of the revisions we have made.

Yours sincerely,

Ray O’Keefe

Response:

Style requirements have been followed according to the recommended guidelines.

Response:

We have now provided two new Excel files (S1 File and S1 Data) with the minimal data set for the study findings.

Reviewers' comments:

Reviewer #1: In this manuscript, carefully prepared by Wood et al., the authors describe their experimental approach to establishing the link between congenital mutations of TXNL4A (a component of core spliceosomal subunit U5 snRNP) and the predominant craniofacial phenotype of the stemming Burn‑McKeown syndrome. The team has prepared induced pluripotent stem cells (iPSCs) of the patient's, patient mother's and unrelated controls' PBMCs and induced its differentiation into neural crest cells (iNCCs). Upon this induction, the authors observed slower proliferation and partly defective differentiation of the patient's iPSCs compared to mother and controls, but not any difference in apoptosis. In both cell types (iPSCs and iNCCs) the authors observed differential expression and splicing when comparing patient and controls (including mother), both of which were more pronounced in the iNCCs. The differentially expressed and spliced genes were enriched in pathways and processes connected to NCC differentiation and function, e.g. the EMT, which is a substantial part of the NCC differentiation. Finally, the authors characterized the subset of exons that were differentially skipped in the patient iNCCs compared to controls and observed some conserved properties among these exons in terms of their length, length of the surrounding introns, etc.

The manuscript is clearly and rigorously written. The methods and experiments are described into sufficient detail, the discussion is a little bit longish but comprehensible and logically evolving. The experiments seem to be rigorously conducted and analysed, well corroborating the findings. I have only minor comments that can be found below.

Response:

The Discussion has been shortened relating to the above comment and the comments of Reviewer 2.

Minor comments:

page 18: „Unlike all other patients with BMKS described to date, the patient displays developmental delay and severe intellectual disability; however, whole exome sequencing has not revealed any genetic variants that could account for this significant cognitive impairment.“

Was the patient tested for structural variants? In any case, the possibility of a structural variant underlying the cognitive impairment ought to be mentioned.

Response:

Yes, structural/copy-number variants were checked for in the patient and nothing was found; she also had a normal karyotype (see reference 7; Strang-Karlsson et al., 2017). Text has been added on line 407 to mention this point.

The text is sometimes a little bit longish. There are a few passages that describe numbers that can be easily observed from a table or a figure, e.g. at page 31, lines 681-684.

Response:

The text has been shortened throughout the Results and Discussion.

Lines 922-923: „However, we did not observe any global changes in splicing factor expression in our BMKS iNCCs.“ As I understand it, the authors did not observe any significant changes in the level of splicing factors expression (except of TXNL4A). However, potentially even splicing changes between patient and controls in a splicing regulator that could be NCC-specific or just important for EMT might be of high relevance, possibly explaining the differences in mis-splicing between iPSCs and iNCC, at least in part. Did the authors observe the results in that way?

Response:

We thank the reviewer for pointing out that we should also be looking at splicing regulators along with core splicing factors. We have now carried out this analysis which is included in S3 Table. We observe some small but significant changes in the splicing of some splicing regulators, especially in the patient iNCCs. We have now described this additional analysis in the text lines 498-501, 538-540, 636-639 and 708-712.

Line 566: one of the control iPCS lines is depicted here as „SW71A“ but as SW171A elsewhere. I guess this is a typing error.

Response:

We thank the reviewer for pointing this typo out, it has now been corrected.

Lines 667-669: „For these genes which were not expressed in either the iPSCs or the iNCCs, it was unclear whether, if the gene had been expressed in both iPSCs and iNCCs, it would have been differentially spliced in both cell types.“ In fact, I do not understand why you mention that. It seems very logic and clear to me, but I see not point in particular commenting that.

Response:

We thank the reviewer for pointing this out, this text has now been removed.

Line 747-748: typping error: „S6 Fig.“ (reverse words order than usually used in the manuscript).

Response:

We have followed the PLOS One formatting here which has the Supplementary info in reverse order.

Line 916: one of the „mis-expressed“ words should be replaced by „mis-spliced“ I guess.

Response:

We thank the reviewer for pointing this typo out, it has now been corrected. 

 

Reviewer #2: Wood and colleagues study a rare craniofacial developmental disorder, the Burn-McKeown syndrome, caused by the 34 base pair deletion in the promoter region of one allele of TXNL4A gene, combined with a loss-of-function variant on the other allele, resulting in reduced TXNL4A expression. TXNL4A gene encodes a component of the U5 snRNP, one of the core components of the spliceosome. The authors hypothesize that the TXNL4A mutations observed in BMKS patients can cause reduced assembly of the human tri-snRNP, thus affecting the splicing of a specific subset of pre-mRNAs. To test this hypothesis, they generate induced pluripotent stem cell (iPSC) lines from a patient with BMKS and her unaffected carrier mother, and from unrelated, unaffected individuals, and characterize the developed models for proliferation, differentiation and gene expression. RNA-Seq analysis revealed transcriptome-wide mis-expression and mis-splicing, in particular aberrant exon skipping, affecting distinct subsets of pre-mRNAs, with an enrichment for genes involved in processes important in NCC specification, highlighting a decreased response to the WNT signaling.

Overall, the experiments are well conducted and the obtained results could be relevant for the potential advances in the rare BMKS disease. However, the manuscript is quite dispersive in the current form and too long; it would benefit of a general shortening, especially in the introduction and discussion sections. Therapeutic potential of the developed findings should be discussed.

Response:

The manuscript has been shortened generally, paying particular attention to the Introduction and Discussion. The therapeutic potential of the findings has been mentioned in the Discussion line 1056.

 A few clarifications and key control experiments are nonetheless required:

 1. The experiments in Figure 3A need to be quantified in some way. Ideally, the authors should perform a morphological analysis by using specific markers of differentiation. Quantification of their signals should be performed.

Response:

The images in Figure 3A were presented to show the qualitative changes in cell morphology associated with iPSC cell differentiation into iNNCs, which was not different between the patient, mother or control cell lines. Specific markers of differentiation are actually presented in Supplementary Figures S5A-D so the visual analysis in Figure 3A is not needed so has been removed.

 2. The GO analyses are not convincing and could be ameliorated by changing the selected parameters, including the EASE score and the graphical representation. In particular, in figure 4A and 4B it seems that each GO term displays the same fold enrichment. P value for each GO term should be added.

Response:

The GO analysis has been carried out according to well established procedures that have been described in detail in the Materials and Methods. All the GO analyses in the manuscript have now been updated to better display the differences in fold enrichment and the P value for each GO term have been added. These are now in Figures 2 and 5. 

 3. Validation of GE changes were performed using the patient sample and the pooled control. Validations should also be performed by taking separately samples from mother and unrelated control.

Response:

Validation of GE changes was performed for the samples using the patient, mother and unrelated controls separately and analysed as such. However, because the RNA-Seq data was analysed comparing patient to pooled mother and unrelated controls, we presented the GE validation using the same pooling for consistency. This point has been made clearer in the text in lines 485-486.

Minor points:

Figure 1A – Western blot analysis to show and quantify TXNL4A protein expression in the derived iPSCs should be performed. In the figure legend please specify that the qPCR analysis is performed to monitor TXNL4A expression.

Response:

We have shown reduced TXNL4A expression at the mRNA level and we see differences in transcriptome and cell behaviour. What is happening at protein level in iPSCs is less important because we expect the manifestations of the variants in TXNL4A to primarily affect the iNCCs, so the pluripotent stage is just a stepping stone to getting there. Additionally, by pooling maternal and control lines for RNA-Seq analysis of both iPSCs and iNCCs, we are reducing background effects so the changes we see are much more likely to derive from changes in TXNL4A expression levels. Finally, the only currently available TXNL4A antibodies are very poor and we would not be confident with any quantitation using these antibodies.

Figure 1A panel and legend have been changed to specify the qPCR analysis was performed to monitor TXNLK4A expression.

Table S3, S6, S7 and S10: It would be suitable to represent the results of the Ingenuity Pathway Analyses by using charts and cluster dendrograms or pathway graphs.

Response:

Yes, we agree that Tables S3, S6, S7 and S10 would be better presented differently. We have now provided heat maps for these Tables which are now presented as Figures S2, S4 and S8.

Table 2 and Table 3 could be represented with graphs or pie charts.

Response:

Yes, we agree that Table 2 and 3 would be better presented as pie charts. We have now provided pie charts for these Tables which are now presented as Figures 3A and 6A.

Response:

All Figures have been checked with PACE and meet the PLOS requirements.

---

## [Decision Letter · Decision Letter 1]

9 Jul 2020

Modelling the developmental spliceosomal craniofacial disorder Burn-McKeown syndrome using induced pluripotent stem cells

PONE-D-20-13089R1

Dear Raymond,

We’re pleased to inform you that your manuscript has been judged scientifically suitable for publication and will be formally accepted for publication once it meets all outstanding technical requirements.

Kind regards,

Emanuele Buratti, Ph.D.

Academic Editor

PLOS ONE

Additional Editor Comments (optional):

Reviewers' comments:

Reviewer's Responses to Questions

**Comments to the Author**

1. If the authors have adequately addressed your comments raised in a previous round of review and you feel that this manuscript is now acceptable for publication, you may indicate that here to bypass the “Comments to the Author” section, enter your conflict of interest statement in the “Confidential to Editor” section, and submit your "Accept" recommendation.

Reviewer #1: All comments have been addressed

Reviewer #2: All comments have been addressed

2. Is the manuscript technically sound, and do the data support the conclusions?

Reviewer #1: Yes

Reviewer #2: (No Response)

3. Has the statistical analysis been performed appropriately and rigorously? 

Reviewer #1: Yes

Reviewer #2: (No Response)

4. Have the authors made all data underlying the findings in their manuscript fully available?

Reviewer #1: Yes

Reviewer #2: (No Response)

5. Is the manuscript presented in an intelligible fashion and written in standard English?

Reviewer #1: Yes

Reviewer #2: (No Response)

6. Review Comments to the Author

Reviewer #1: The authors addressed all my comments to my full satisfaction. The manuscript is ready for publication.

Reviewer #2: (No Response)

7. PLOS authors have the option to publish the peer review history of their article (what does this mean?). If published, this will include your full peer review and any attached files.

Reviewer #1: **Yes: **Lucie Grodecká

Reviewer #2: No

---

## [Editor Report · Acceptance letter]

17 Jul 2020

PONE-D-20-13089R1 

Modelling the developmental spliceosomal craniofacial disorder Burn-McKeown syndrome using induced pluripotent stem cells 

Dear Dr. O'Keefe:

I'm pleased to inform you that your manuscript has been deemed suitable for publication in PLOS ONE. Congratulations! Your manuscript is now with our production department. 

Kind regards, 

on behalf of

Dr. Emanuele Buratti 

Academic Editor

PLOS ONE